# ENHANCING CHAIN-OF-THOUGHTS PROMPTING WITH ITERATIVE BOOTSTRAPPING IN LARGE LANGUAGE MODELS

## ABSTRACT

Large language models (LLMs) can achieve impressive performance on various reasoning tasks by incorporating chain-of-thought (CoT) prompting, where step-by-step reasoning is provided to guide LLMs to generate answers to questions, and the question-rationale-answer triplets are utilized as demonstration exemplars. However, the reasoning chains of demonstrations generated by LLMs are observed to be prone to errors, which can subsequently lead to incorrect reasoning during inference. Furthermore, inappropriate exemplars, e.g., overly simplistic or complex exemplars depending on the question's difficulty level, can affect the LLM's performance. To address these issues, we introduce Iter-CoT (**Ite**rative boots**t**rapping in **C**hain-**o**f-**T**houghts prompting). Iter-CoT has two advantages: (1) it adopts iterative bootstrapping that enables LLMs to rectify errors autonomously, resulting in more precise and comprehensive reasoning chains. (2) it selects exemplars of challenging yet answerable (i.e., the LLM has the potential to answer correctly) questions, enhancing the LLMs' generalizability to answer questions with varying difficulty levels. Experimental results exhibit Iter-CoT superior performance on three distinct reasoning tasks on ten datasets.

## 1 INTRODUCTION

Chain-of-Thought (CoT) (Wei et al., 2022) prompting is a technique to enhance the reasoning abilities of Large language models (LLMs) by generating a series of reasoning steps to obtain the answer, and the reasoning chains are utilized as exemplars to demonstrate the task and provide In-Context Learning (ICL) (Brown et al., 2020) to LLMs. Recently, LLMs (Chowdhery et al., 2022; Thoppilan et al., 2022; Rae et al., 2021; Smith et al., 2022; Scao et al., 2022) with CoT prompting have demonstrated remarkable performance in complex reasoning tasks, including arithmetic (Cobbe et al., 2021; Ling et al., 2017; Hosseini et al., 2014; Koncel-Kedziorski et al., 2015; Patel et al., 2021; Miao et al., 2020), commonsense (Talmor et al., 2019; Geva et al., 2021; Kojima et al., 2022; Wei et al., 2022), and symbolic reasoning (Wei et al., 2022).

Existing studies on CoT prompting can be classified into two categories. The first category is manually constructed CoT prompting((Wei et al., 2022; Diao et al., 2023; Fu et al., 2022; Wang et al., 2022; Li et al., 2022; Zhou et al., 2022; Press et al., 2022)), where human annotators manually craft a collection of question-rationale-answer exemplars to guide the model's reasoning process. However, human annotations' inherent subjectivity and limitations make these approaches costly, sub-optimal, and highly inconsistent. The second category is automatically generated CoT prompting((Kojima et al., 2022; Zhang et al., 2022; Shum et al., 2023; Shao et al., 2023)), where LLM-generated CoT is utilized. In practice, reasoning chains generated by LLMs have demonstrated superior performance compared with human annotations.

However, three issues remain under-explored in the literature. (1) **Difficulty of questions**. It is regarded that questions of mediate difficulty level can best guide the LLMs (Diao et al., 2023). As shown in Figure 1, it is observed that simplistic examples (Simple-CoT) perform poorly in solving complex questions of more hops[1], while excessively complex exemplars (Complex-CoT (Fu et al.,

---

[1]Following (Shum et al., 2023; Fu et al., 2022), we measure the question's difficulty by the number of hops in the rationale, with fewer hops indicating simpler questions and more hops indicating more complex questions.

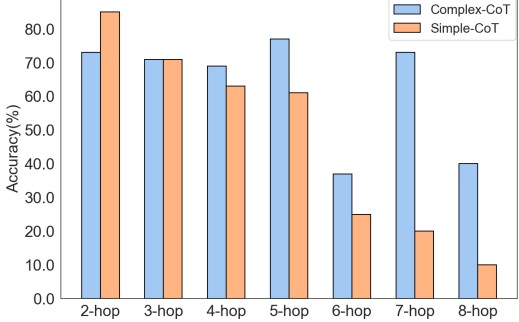 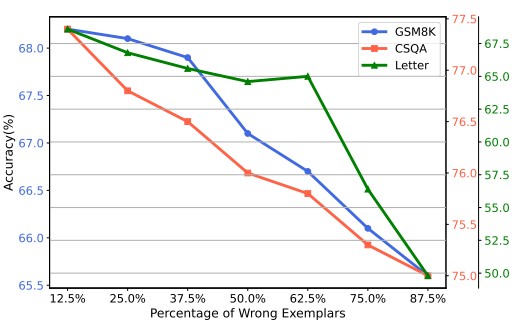

Figure 1: Effect of different demonstrations (Simple-CoT v.s., Complex-CoT) on different questions (difficulty from 2-hop to 9-hop) on GSM8K dataset.

Figure 2: Impact of wrong exemplars on three different benchmarks (GSM8K, CSQA and Letter).

2022)) are unsatisfying on simpler questions. (2) **Correctness of demonstration**. Reasoning chains of demonstrations generated by LLMs are prone to errors (Zhang et al., 2022; Diao et al., 2023), which can significantly reduce overall performance. As shown in Figure 2, accuracy on various datasets decreases as incorrect exemplars increase. (3) **Missing contextual information**. Previous works merely combine the question and the "let's think step by step" prompt (Kojima et al., 2022) during the generation of demonstrations without incorporating contextual information (such as incorrect reasoning chains and feedback answers generated by LLMs). Missing contextual information limits the LLM's capability to learn from previous reasoning errors and avoid making similar errors.

In order to address the issues above, we propose Iter-CoT (**Ite**rative bootst**r**apping in **C**hain-**o**f-**T**houghts Prompting). Iter-CoT allows LLMs to self-correct and summarize the more precise and comprehensive reasoning chains, which identify challenging yet answerable (i.e., LLM has the potential to answer correctly) questions as demonstrations in order to enhance the LLMs' generalizability to answer questions with varying difficulty levels. We evaluate Iter-CoT on three distinct reasoning tasks (arithmetic, commonsense, and symbolic) across ten datasets. The experimental results show that Iter-CoT significantly outperforms existing prompting approaches.

Our contributions are summarized as follows: (1) We propose a new paradigm for CoT, which generates precise and comprehensive reasoning chains during interaction with LLMs. To our knowledge, our work is the first to illustrate the importance of iterative interaction with the LLMs to generate high-quality demonstrations. (2) We propose Iter-CoT, an approach that generates self-corrected and summarized reasoning chains on exemplars with intermediate difficulty levels, which are utilized as demonstrations to enhance the LLMs' performance. (3) We implement Iter-CoT under both labeled and unlabeled conditions, achieving **state-of-the-art (SOTA)** results in both scenarios across ten datasets within three distinct tasks.

## 2 MOTIVATION

We propose Iter-CoT, which enhances LLMs' reasoning performance by integrating iterative bootstrapping to self-correct the reasoning chains in demonstrations.

### 2.1 THE SELF-CORRECTION ABILITY OF LLMS

LLMs have the potential to self-correct. Wang et al. (2022) demonstrated the ability of LLMs to generate multiple diverse answers for the same question. We conduct an empirical experiment on the GSM8K dataset to show that LLMs have the potential to generate the correct reasoning chain for questions that are initially answered erroneously. The GSM8K dataset contains groundtruth answers for 7473 questions in training set. For each question in training set, we first apply the zero-shot-CoT to generate answers. For questions that are answered incorrectly, we prompt the LLM with a hint "the answer is incorrect" to re-answer. The process is repeated for six iterations until the number of correctly answered questions no longer increases.

As shown in Figure 3), the performance of the LLMs is improved (i.e., increasing from 4089 (54.7%) to 4898 (59.1%) after the first iteration, and the improvement sustains in subsequent iterations, ultimately reaching a peak of 5726 (76.6%). This observation suggests that LLMs can autonomously rectify errors with hints and contextual information.

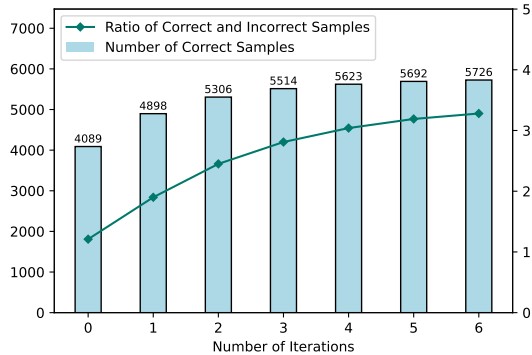

Figure 3: Effect of re-answering the question based on the hint and previous rationales.

## 2.2 THE VALUE OF REVISED EXAMPLES

Examples containing erroneous rationales were ignored or screened out to prevent their adverse effects in previous studies (Zhang et al., 2022; Shum et al., 2023). However, inspired by the idea that students can improve their problem-solving abilities by learning from a collection of mistakes, we believe that allowing the model to learn from examples that have been answered incorrectly and then corrected can also effectively enhance the model's inference performance. Figure 4 is a case study of the value of revised examples on the Date Understanding (Wei et al., 2022) dataset. It shows the different effects of two distinct demonstrations on the same test question. Using a randomly sampled exemplar as a demonstration is not beneficial (right side), even though it is already correctly answered. On the contrary, using the revised example's reasoning chain (left side), where the reasoning chain is self-corrected by the LLM with the approach in Section 2.1, improves LLM's reasoning ability.

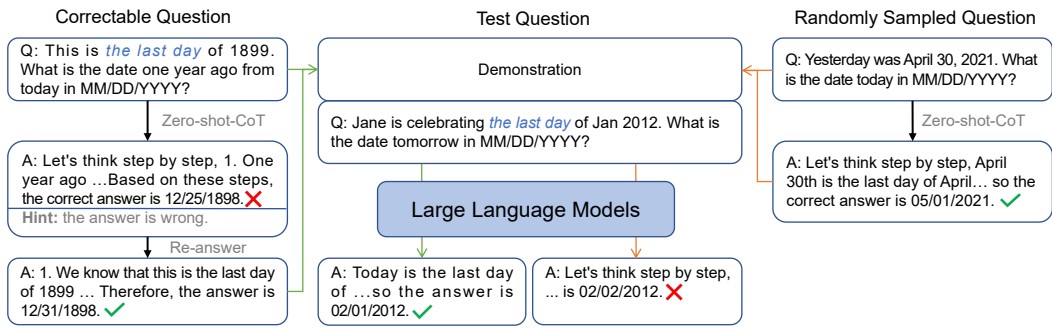

Figure 4: The illustration of the value of revised examples. Challenging yet answerable exemplars as demonstrations can enhance the model's reasoning performance.

## 3 ITER-COT: ITERATIVE BOOTSTRAPPING IN CHAIN-OF-THOUGHT PROMPTING

Motivated by the observations in Section 2, we propose Iter-CoT (**Ite**rative boots**tr**apping in **C**hain-**o**f-**T**houghts prompting), which generates the chain-of-thought demonstrations by guiding the LLM to rectify errors and summarize the reasoning chains on questions with appropriate difficulty levels. Following that, we put these exemplars into the demonstration pool. During the inference, we sampling and fixed the exemplars as the demonstration.

As shown in Figure 5, Iter-CoT consists of two stages, the **construction** stage of the demonstration pool and the **inference** stage. Moreover, the construction of the demonstration pool consists of three phases:

- **Initialization**. The Zero-Shot-CoT (Kojima et al., 2022) method is employed on the training set to prompt the LLM to generate reasoning chains and answers. Error examples are recorded for the subsequent phases.

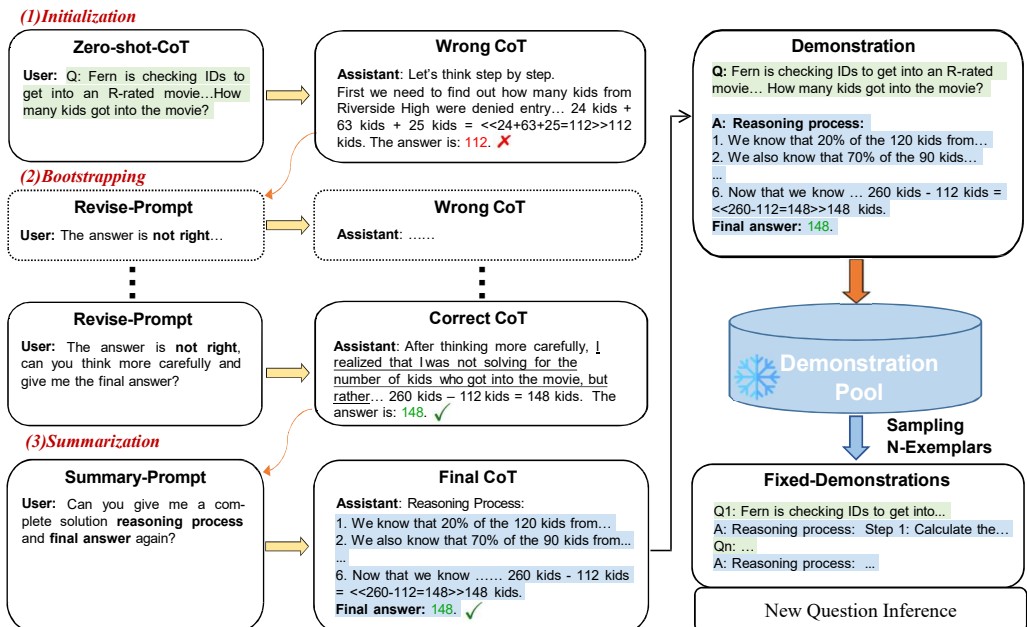

Figure 5: The workflow of Iter-CoT: **1. The construction of the demonstration pool**: 1) *Initialization*: we query the LLMs to generate reasoning chain and answer with Zero-Shot-CoT (Kojima et al., 2022). 2) *Bootstrapping*: we use `Revise-Prompt` to guide LLMs to revise the reasoning chain repeatedly until the generated CoT is completely accurate. 3) *Summarization*: we prompt LLMs with `Summary-Prompt` to generate the final reasoning chain(refered as Final CoT) based on the contextual information provided within the overall process. Then we add the Final CoT with the corresponding question as an example to the demonstration pool. **2. Inference**: LLMs generate answers for the test questions with the demonstrations sampled from the constructed demonstration pool.

- **Bootstrapping**. For each erroneous example, the `Revise-Prompt` ("Your answer is not right; can you think more carefully and give me the final answer?") is utilized to guide the LLM to self-correct until the correct answer is generated. In the absence of a prompted reference answer, the correct answer often corresponds to the correct reasoning chains.

- **Summarization**. Once the correct answers are obtained, the `Summary-Prompt` ("Can you give me a complete solution reasoning process and final answer again?") is employed to guide the LLM in reviewing the previous rationales and summarizing the final reasoning chains. This process enables the LLM to capture rich contextual information from multi-turn conversations, resulting in more precise and comprehensive reasoning chains.

Upon completion of the aforementioned process, the final generated reasoning chain is combined with the corresponding question and added to the demonstration pool until the sample size fulfills the requirements. The requirement is flexible, yet at least satisfies differences in various datasets shown in Table 6.

Our approach works in both label-available and non-available scenarios. We use a rule-based approach to determine the correctness of the answers when labels are available in the construction stage of the demonstration pool. In contrast, when labels are unavailable, we use a more powerful model (e.g., GPT-4 (OpenAI, 2023)) as an evaluator to assess the correctness of the answers.

**Inference**: During the inference stage, a random sampling approach is used to select $N$ exemplars from the demonstration pool, which served as fixed demonstrations for the entire test set.

## 4 EXPERIMENT

### 4.1 DATASETS AND EVALUATION METRICS

We evaluate our methods on ten datasets across three categories of different reasoning tasks, including (1) six arithmetic reasoning datasets: GSM8k (Cobbe et al., 2021), AQuA (Ling et al., 2017), AddSub (Hosseini et al., 2014), SingleEq (Koncel-Kedziorski et al., 2015), SVAMP (Patel et al., 2021) and ASDiv (Miao et al., 2020); (2) three commonsense reasoning datasets: CSQA (Talmor et al., 2019), StrategyQA (Geva et al., 2021) and Date Understanding (Wei et al., 2022); (3) one symbolic reasoning datasets: Letter Concatenation (Wei et al., 2022). Examples of each reasoning task and a detailed description of each dataset are shown in Table 6 and Table 7.

In the inference stage, we report the exact match accuracy as our evaluation metric following previous works (Wei et al., 2022; Kojima et al., 2022).

### 4.2 BASELINES

We compare our methods with five baseline approaches: **Manual-CoT** (Wei et al., 2022), **Random-CoT**, **Complex-CoT** (Fu et al., 2022), **Auto-CoT** (Zhang et al., 2022) and **Self-Consistency (SC)** (Wang et al., 2022). Manual-CoT involves using manually constructed reasoning chains as exemplars, listed in the appendix of Wei et al. (2022). Random-CoT randomly selects $n$ questions from the training set and generates chains using the "let's think step by step" prompt. Complex-CoT selects most complex exemplars, such as exemplars with most complex rationales or longest questions from the training set, as demonstrations. Auto-CoT utilizes clustering techniques to sample questions and generate chains with the same approach. Specifically, we implement Auto-CoT by generating reasoning chains for the questions provided in their appendix as demonstrations. Self-Consistency generates multiple answers for a question and uses a majority voting mechanism to select the final answer.

### 4.3 IMPLEMENTATION DETAILS

We implement Iter-CoT on GPT-3.5-turbo (Ouyang et al., 2022) and GPT-4 (OpenAI, 2023), using the OpenAI API[2]. We implement Iter-CoT on open-source models using 8 A100-40Gs for inference using Llama-2-70B-Chat and Llama-2-70B (Touvron et al., 2023) without quantization in our experiments. During the construction stage of the demonstration pool, we utilize a temperature setting of 0.7, whereas during the inference stage, we fix the temperature to 0 for reproducibility. Moreover, we set temperature = 0.7 and n = 40 for evaluation under self-consistency as (Wang et al., 2022). We adopt the number of exemplars for each dataset based on the experimental configuration of prior work (Zhang et al., 2022; Fu et al., 2022). Specifically, for datasets lacking test sets and without comparable datasets for transfer (e.g., Date Understanding), we randomly select a small portion as the training set and reserve the remaining portion for evaluation as the test set. In addition, We conducted three trials and averages for each experiment requiring random sampling to obtain final results. The size of each dataset and the partitioning of train and test sets are shown in Table 6.

### 4.4 MAIN RESULTS

As Iter-CoT can be applied with and without groundtruth labels, we implement two versions: Iter-CoT(w/ label) and Iter-CoT(w/o label). The latter is implemented with GPT-4 as the evaluator. The experimental results are presented in Table 1. We have the following observations:

**Iter-CoT achieves superior performance on different tasks.** Without using annotations/labels, Iter-CoT achieves superior performance on different tasks, and its performance is comparable or even superior than methods with annotations/labels. When examining the results on the first five arithmetic reasoning tasks in Table 1, we observe that Iter-CoT(w/o label) outperforms Complex-CoT with its average scores surpassing those of Complex-CoT 0.4%. When labels are available, Iter-CoT can achieves the best average score(81.5%) on all ten datasets across three tasks with GPT-3.5-turbo, surpassing Manual-CoT by 3.8% and Random-CoT by 6.1%. Notably, on the Letter Concatenation

---

[2]https://platform.openai.com/

| Method | Annotation/Label Needed | Arithmetic | | | | | | Commonsense | | | Symbolic | Avg |
|---|---|---|---|---|---|---|---|---|---|---|---|---|
| | | GSM8K | AQuA | AddSub | SingleEq | SVAMP | ASDiv | CSQA | STQA | Date | Letter | |
| *UL2-20B** | Annotation | 4.4 | 23.6 | 18.2 | 20.2 | 12.5 | 16.9 | 51.4 | 53.3 | - | 0.0 | - |
| *LaMDA-137B** | Annotation | 14.3 | 20.6 | 51.9 | 58.7 | 37.5 | 46.6 | 57.9 | 65.4 | - | 13.5 | - |
| *PaLM-540B** | Annotation | 56.9 | 35.8 | 91.9 | 92.3 | 79.0 | 73.9 | 79.9 | 77.8 | - | 63.0 | - |
| *GPT-3.5-turbo* | | | | | | | | | | | | |
| Random-CoT | No | 72.6 | 53.8 | 89.9 | 95.9 | 82.0 | 88.6 | 74.8 | 58.7 | 64.5 | 73.2 | 75.4 |
| Auto-CoT | No | 78.1 | 56.7 | 94.7 | 96.4 | 83.6 | - | 72.3 | 62.8 | - | 78.2 | - |
| **Iter-CoT(w/o label)** | No | 80.5 | 58.7 | 92.7 | **97.2** | **85.0** | 90.4 | **76.1** | 63.5 | 78.3 | **88.6** | 81.1 |
| Manual-CoT | Annotation | 74.9 | 55.5 | 93.4 | 96.4 | 82.4 | 89.5 | 75.0 | **66.1** | 70.0 | 74.2 | 77.7 |
| Complex-CoT | Annotation | **82.0** | 57.4 | 93.2 | 96.5 | 81.9 | - | - | - | - | - | - |
| **Iter-CoT(w/ label)** | Label | 80.9 | **62.2** | **94.9** | 96.9 | 84.3 | **91.0** | 75.9 | 64.5 | **78.6** | 85.2 | **81.5** |
| Manual-CoT-SC | Annotation | 80.8 | 60.6 | 94.2 | 96.6 | 82.7 | 89.6 | 80.1 | **67.8** | 73.0 | 78.6 | 80.4 |
| **Iter-CoT(w/o label)-SC** | No | 86.8 | 69.2 | 94.4 | **97.8** | 84.7 | **91.8** | 79.5 | 64.3 | **82.1** | 88.1 | 83.9 |
| **Iter-CoT(w/ label)-SC** | Label | **89.1** | **72.4** | **94.9** | 97.3 | **85.2** | 91.2 | **80.6** | 66.7 | 80.7 | **89.6** | **84.8** |

Table 1: Accuracy on ten datasets from arithmetic, commonsense and symbolic reasoning tasks. * denotes all three LLMs use Manual-CoT. The content in the "Annotation/Label Needed" column indicates whether the corresponding method requires annotation of the complete reasoning chain or label of the final answer. Iter-CoT(w/o label) is implemented with GPT-4 as the evaluator. The best results without Self-Consistency (SC) on GPT-3.5-turbo are highlighted with green color, and the best results with SC on GPT-3.5-turbo are highlighted with blue color.

dataset, Iter-CoT(w/ label) achieved remarkable improvements of 7% compared to the previous highest scores. On the first five arithmetic reasoning tasks in Table 1, Iter-CoT (w/ labels) continues to exhibit the best performance, at 83.8%, surpassing Complex-CoT, where the annotations of reasoning chains in demonstrations are needed, by 1.6%. In conclusion, our approach outperforms existing approaches and achieves **state-of-the-art** results across various tasks.

**Iter-CoT(w/o label) share comparable performance with Iter-CoT(w/ label).** Iter-CoT(w/o label) demonstrates a marginal superiority over Iter-CoT(w/ label) on the Singleeq and SVAMP datasets, with improvements of 0.3% and 0.7%, respectively. However, it registers slightly lower performance compared to Iter-CoT(w/ label) on all other datasets, resulting in the average score that is 0.6% lower than that of Iter-CoT(w/ label). The performance gap between Iter-CoT(w/o label) and Iter-CoT(w/ label) can be attributed to the inherent challenge of using GPT-4 for evaluating the correctness of responses. The errors generated during the evaluation would cause the selected demonstrations to be answered incorrectly initially or not with the correct reasoning chains. Nonetheless, the impact of these errors on the overall results remains acceptable. Statistically, GPT-4 demonstrates an 87.5% accuracy in determining the correctness of responses during the demonstration pool construction stage. Consequently, the proportion of non-compliant samples in the final selected demonstrations remains acceptable. Furthermore, the incorrectly evaluated demonstrations tend to be challenging, thereby offering valuable insights to LLMs.

**Self-consistency (SC) consistently augments the efficacy of all methodologies.** Notably, on the GSM8K and AQuA datasets, SC significantly improves model inference performance, resulting in respective enhancements of 5.9%, 7.2%, and 8.2% for the Manual-CoT, Iter-CoT(w/o label), and Iter-CoT(w/ label) methods on GSM8K, and 5.1%, 10.5%, and 10.2% on AQuA. On other datasets, SC has also demonstrated consistent improvements. Ultimately, across the ten datasets, the three methods exhibit average performance enhancements of 2.7%, 3.1%, and 3.4%, respectively. Moreover, With the inclusion of SC, Iter-CoT(w/ label) and Iter-CoT(w/o label) continue to exhibit consistent superiority over Manual-CoT, with an average score advantage of 4.4% and 3.5%, respectively.

## 4.5 Performance on Different Foundation Models

In order to validate the feasibility of our approach across various diverse models, we conduct experiments on GPT-4 and two open source models: Llama-2-70B and Llama-2-70B-Chat (Touvron et al., 2023), as shown in Table 2. When comparing the results of different methods within three distinct foundation models, we observe that our approach consistently outperforms Manual-CoT across varying models. Specifically, on the Llama-2-70B-Chat, Llama-2-70B, and GPT-4 models, the

average improvement of Iter-CoT(w/ label) over Manual-CoT is 14.9%, 7.6%, and 3.6%, respectively. Furthermore, Iter-CoT(w/o label) exhibits performance closely aligned with Iter-CoT(w/ label) across diverse models, with an average score difference of merely 2.1%, 1%, and 2.5% within the three models.

This demonstrates the robustness of our proposed methods: there is a stable improvement for different foundation models.

| Method | GSM8K | CSQA | Date | Letter | Avg. |
|---|---|---|---|---|---|
| *Llama-2-70B-Chat* | | | | | |
| Manual-CoT | 50.7 | 69.6 | 42.3 | 22.6 | 46.3 |
| Iter-CoT(w/o label) | 58.2 | 66.2 | 65.3 | 46.7 | 59.1 |
| Iter-CoT(w/ label) | 59.1 | 67.6 | 68.2 | 49.8 | 61.2 |
| *Llama-2-70B* | | | | | |
| Manual-CoT | 56.8 | 68.4 | 73.3 | 22.4 | 55.2 |
| Iter-CoT(w/o label) | 61.1 | 73.1 | 75.6 | 37.2 | 61.8 |
| Iter-CoT(w/ label) | 62.3 | 71.1 | 77.3 | 40.6 | 62.8 |
| *GPT-4* | | | | | |
| Manual-CoT | 92.0 | 83.0 | 90.1 | 92.9 | 89.5 |
| Iter-CoT(w/o label) | 94.3 | 83.5 | 93.5 | 95.1 | 91.6 |
| Iter-CoT(w/ label) | 95.2 | 85.7 | 94.7 | 96.6 | 93.1 |

Table 2: Different Approaches' Performance with Llama-2-70B-Chat, Llama-2-70B and GPT-4 on Four Datasets. Iter-CoT(w/o label) is implemented with GPT-4 as the evaluator.

| Method | GSM8K | Date | Letter | Avg. |
|---|---|---|---|---|
| Init-correct-CoT | 72.5 | 63.0 | 78.6 | 71.4 |
| Random-CoT | 72.6 | 64.5 | 73.2 | 70.1 |
| Correct-CoT | 79.2 | 67.0 | 82.6 | 76.3 |
| Best-of-N-CoT | 76.3 | 66.7 | 77.4 | 73.5 |
| Iter-CoT | 80.9 | 71.3 | 85.2 | 79.1 |

Table 3: Accuracy with different methods on GPT-3.5-turbo. *Init-correct-CoT*: just take the correct exemplars after initialization; *Random-CoT*: randomly sample exemplars after initialization; *Correct-CoT*: Iter-CoT without summarization step; *Best-of-N-CoT*: prompt the LLM to generate multiple responses and choose the correct one as the exemplar; *Iter-CoT*: Our method with label.

## 4.6 ABLATION STUDIES

During the construction stage of the demonstration pool, both bootstrapping and summarization phases play crucial roles in generating the final exemplars. We conduct a series of ablation experiments to investigate the impact of these two phases on the results. Explanations for all the methods employed in this section of ablation experiments can be found in the caption of Table 3.

In addition, for Iter-CoT(w/o label), we also investigate the accuracy of LLM evaluators and its impact on the results.

### 4.6.1 IMPACT OF BOOTSTRAPPING AND SUMMARIZATION PHASE

**Comparison of Methods with and without a Bootstrapping Phase**    To investigate the impact of bootstrapping phase on model's performance, We contrast Correct-CoT, the method only uses the bootstrapping process, with many methods that do not modify the reasoning chains when generating demonstrations. These methods include Random-CoT, Init-Correct-CoT and Best-of-n-CoT. From Table 3, we observe that the average score for Random-CoT is the lowest, at 70.1%, primarily because Random-CoT may include many erroneous exemplars. Following this is the Init-correct-CoT, with an average score of 71.4%, which is attributed to the inclusion of exemplars where questions were initially answered correctly, thus making them overly simplistic and offering limited assistance to the model. The Best-of-N-CoT achieves the second-highest average score, at 73.5%, as it may also select exemplars containing questions initially answered correctly. The Correct-CoT attains the highest average score, standing at 76.3%, driven by the exemplars generated during the revision phase. These exemplars feature questions initially answered incorrectly by the model but subsequently corrected through guidance. Such questions pose a certain level of challenge to the model while also offering the potential for correct answers. Therefore, the revision phase can, to the greatest extent possible, ensure the correctness of exemplars while selecting questions of appropriate difficulty.

**The Effect of Iterations in the Bootstrapping Phase**    We posit that questions requiring multiple times of bootstrapping are more challenging. Consequently, we conduct experiments for Iter-CoT on three datasets with iterative bootstrapping, where the iterations are the number of times the bootstrapping phase is invoked. The results are depicted in Figure 6. We observe that with increasing iterations, different implementations of Iter-CoT exhibited varying performances across different

datasets. For instance, on the Letter Concatenation, the performance of Iter-CoT (w/ label) consistently declines with iterations, while Iter-CoT (w/o label) shows a continuous improvement.

Iter-CoT displays a trend of initially rising and then falling accuracy on several datasets. However, even as accuracy decreases with increasing iterations, the post-decline accuracy still outperforms most baselines. This indicates that Iter-CoT, through iterations, can identify the most suitable examples within a certain range. In Table 1, for GSM8K, CSQA, Letter Concatenation and other datasets sharing the same exemplars with GSM8K (AddSub, SingleEq, SVAMP, and ASDiv), we utilize the best exemplars in this section. In future work, further iterations can be explored to generate exemplars of higher quality for other datasets.

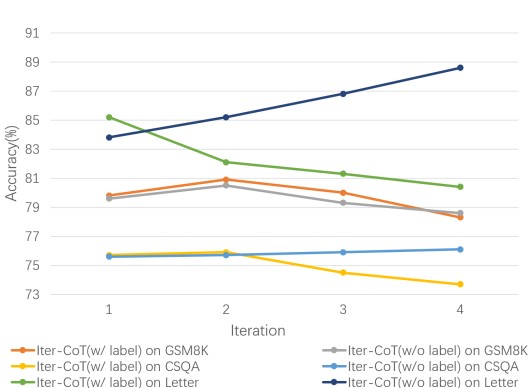

Figure 6: The Iter-CoT's overall performance with iterative bootstrapping on three datasets.

**Impact of Summarization phase**   To investigate the impact of the summarization phase on the model's inference capability, similar to the previous section, we compare Iter-CoT with Correct-CoT, which excludes the utilization of the summarization phase. Table 3 demonstrates that the former outperforms the latter by 2.8%. We attribute this performance difference to the role of the summarization phase, which encourages the model to incorporate extensive contextual information, thereby facilitating the generation of more intricate and comprehensive reasoning chains.

### 4.6.2   IMPACT OF LLM EVALUATORS' ACCURACY IN ITER-COT

In Section 4.4, we mentioned the potential errors when using GPT-4 as an evaluator, which could impact the results. To investigate the influence of evaluators' accuracy on model inference capability in Iter-CoT(w/o label), we select three distinct LLMs and employ them as both the foundation model and evaluator. For each experiment, we compute the evaluator's accuracy along with the final inference performance of the foundation model. The experimental results are presented in Figure 7.

All three subplots exhibit a common trend: the performance of the foundation model improves as the evaluator's accuracy increases. It is consistent with our prior analysis. As the evaluator's accuracy rises, the quality of generated exemplars is close to that of Iter-CoT (w/ label). Furthermore, by comparing the three subplots, we observe that the evaluator tends to achieve higher accuracy in judging the generated answers of weaker foundation models. For example, utilizing GPT-4 as an evaluator to assess the accuracy of Llama-2-70B-Chat yields a precision of 89.2%. In contrast, when

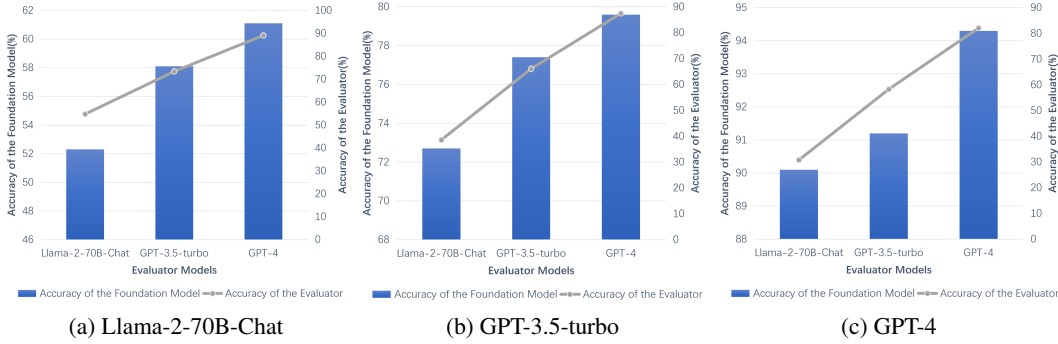

(a) Llama-2-70B-Chat            (b) GPT-3.5-turbo            (c) GPT-4

Figure 7: The influence of evaluator accuracy on model inference performance. Each subfigure corresponds to a foundation model and three evaluators.

evaluated by ChatGPT, the accuracy stands at 73.4%. Llama's self-assessment, however, indicates a modest 54.8% accuracy.

Due to the page limit, experiments and analysis on performance across different levels of difficulty, the impact of using different numbers of seed examples during inference, the comparison between Iter-CoT and CoT, and the length of generated reasoning chains are not included in this section. Details of these experiments can be found in Section B.1, B.2, B.3 and B.5.

## 5 RELATED WORK

### 5.1 CHAIN-OF-THOUGHT PROMPTING

#### 5.1.1 MANUALLY CONSTRUCTED CoT PROMPTS

Wei et al. (2022) proposed Manual-CoT, an approach that employs manually-crafted demonstrations as prompts. In subsequent work, Wang et al. (2022) introduced a novel decoding strategy "Self-Consistency", which generates multiple answers from LLMs and aggregates them through a majority voting mechanism. Li et al. (2022) increased the randomness of the prompts to enhance the diversity of generated reasoning paths. Diao et al. (2023) annotated the reasoning chain manually for the most uncertain questions. Although these approaches have shown remarkable performance in enhancing the model's reasoning capability, they are expensive, suboptimal and highly sensitive.

#### 5.1.2 AUTOMATICALLY GENERATED CoT PROMPTS

Kojima et al. (2022) proposed a "Let's think step by step" prompt that guides LLMs to generate reasoning steps without manually constructed demonstrations. Following this work, Zhang et al. (2022) and Shum et al. (2023) employed zero-shot-cotKojima et al. (2022) to generate the reasoning process in its demonstration. In contrast, Shao et al. (2023) employed seed demonstrations to synthesize more examples by automatically repeating forward and backward processes.

We propose a novel approach to generate reasoning chains by allowing LLMs to retrace their reasoning process after inferring the answer.

### 5.2 IN-CONTEXT LEARNING

In-Context Learning (ICL) is a technique that allows LLMs to complete target tasks during inference by using a few tasks-specific examples as demonstrations, without modifying the model parameters (Shao et al., 2023; Brown et al., 2020). Zhao et al. (2021) underscored that the accuracy of LLMs in ICL depends heavily on the selection and permutation of exemplars. Therefore, significant efforts have been invested in developing approaches to select appropriate few-shot demonstrations.

Zhang et al. (2022) adopted a clustering-based method to select demonstrations. Fu et al. (2022) selected the demonstrations with the most reasoning steps. Similarly, Diao et al. (2023) chose the demonstrations with most uncertain questions. Additionally, Shum et al. (2023) added the demonstrations with the correct answer to the samples pool and sampled the exemplars with a trained model. These studies all strive to minimize the use of incorrect exemplars. Contrarily, Zelikman et al. (2022) handled erroneous examples by hinting the model with the correct answers to generate results again. We conduct a comparative analysis with their approach, which is presented in Section B.4.

Through iterative bootstrapping, our approach selects challenging yet answerable exemplars, enhancing the LLMs' generalizability across varying difficulty levels.

## 6 CONCLUSION

This paper proposes Iter-CoT, an iterative bootstrapping in chain-of-thoughts prompting for large language model reasoning. Unlike previous work, our method prompts LLMs to self-correct their errors in reasoning chains by leveraging iterative bootstrapping and obtaining more precise and comprehensive reasoning chains. Experimental results on ten reasoning datasets among three different reasoning tasks demonstrate that our approach significantly outperforms the previous methods, achieving new state-of-the-art.

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

# A   LIMITATIONS AND FUTURE WORK

**Limitations**   In Iter-CoT(w/o label), the model's performance relies significantly on the accuracy of the evaluator. Therefore, a more robust model than the primary one is required to serve as the evaluator. Moreover, the introduction of the evaluator also results in increased overhead.

**Future Work**   The tasks employed in our present experimental setup involve clear answers, such as GSM8K that the final answer can be used to determine whether the response is correct. In future research, we can explore applying Iter-CoT (w/o label) to tasks with non-unique answers, such as summary generation.

# B   ANALYSIS FOR ITER-COT

## B.1   PERFORMANCE ACROSS DIFFERENT LEVELS OF DIFFICULTY

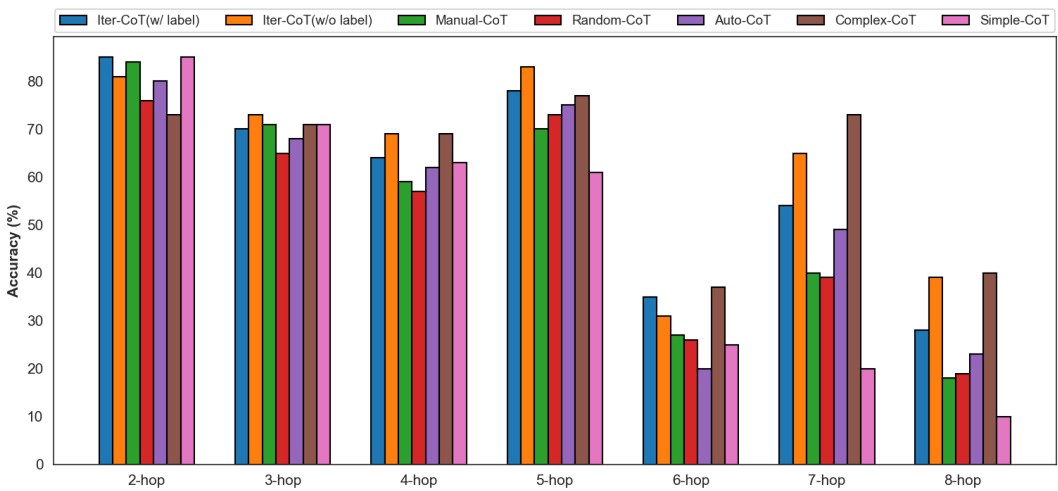

Figure 8: The performance on GSM8K across different numbers of hops.

We investigate the generalization ability of Iter-CoT on questions with varying difficulty levels. We follow the same hop-based criterion as previous work (Fu et al., 2022) to measure query difficulty. We sort the test set of GSM8K according to the number of hops of the annotated reasoning chains and conduct experiments using Iter-CoT and other baselines, as shown in Figure 8. Our results indicate that Iter-CoT is comparable to other methods for questions with few hops, whereas its performance is significantly better than other methods for questions with more hops. Iter-CoT performs the same as Simple-CoT on 2-hop questions (1% higher), while it is on par with Complex-CoT on 8-hop questions (2% lower) and is substantially ahead of the other methods (about 20%). This suggests that with Iter-CoT, we can select exemplars with intermediate difficulty levels, which can greatly improve the performance of LLM among questions of varying difficulty. Moreover, the effect of Iter-CoT(w/o label) is even superior to Iter-CoT(w/ label) on 7-hop and 8-hop questions, which shows the robustness of our proposed methods.

## B.2   EFFECTIVE OF DIFFERENT NUMBERS OF SEED EXAMPLES

In order to investigate the sensitivity of our approaches and conventional CoT to the seed examples, we conducted an experiment on the GSM8K dataset as shown in Figure 9. It demonstrates that both of our approaches outperform CoT, and are more stable as the number of examples increases. Additionally, our experiment also shows that the overall performance is not determined by the quantity of increasing exemplars. For instance, the Iter-CoT peak occurs at five exemplars, while the Iter-CoT(w/o label) and Random-CoT peaks at four exemplars.

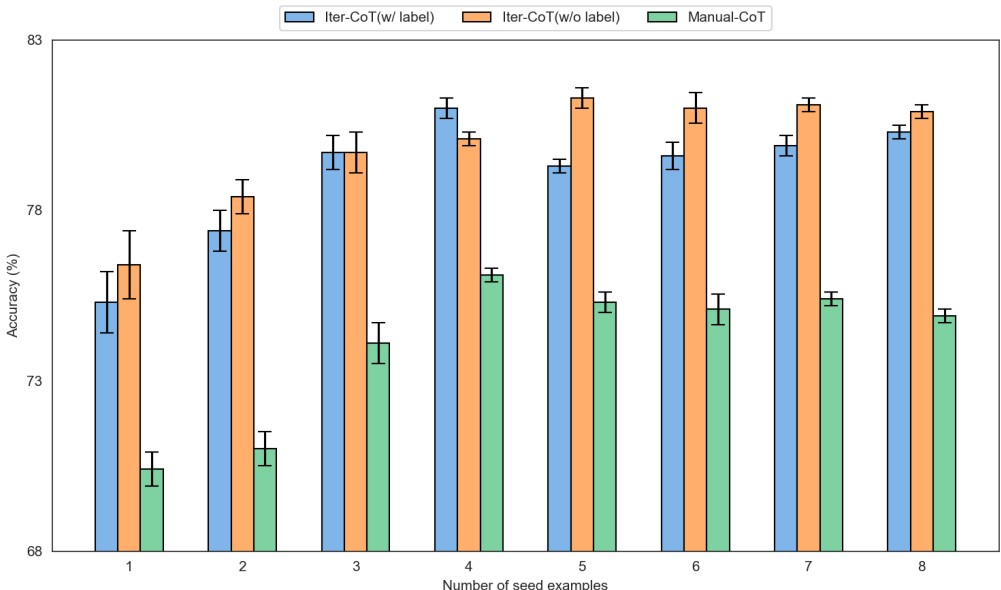

Figure 9: Effective of Different Numbers of Seed Examples on GSM8K.

### B.3 COMPARISON WITH COT

As Section 3 mentions, Iter-CoT can generate more precise and comprehensive reasoning chains than zero-shot-CoT. We conduct inference on three distinct reasoning datasets (GSM8K, Letter(4) and Date Understanding) utilizing both Iter-CoT's first stage and Zero-Shot-CoT with the same questions, which are shown in Table 4. We use the same LLMs and temperature to generate reasoning chains and answers. We observe that the Final CoT generated after the Iter-CoT's first stage is naturally more precise and comprehensive compared to CoT generated by zero-shot-CoT, resulting in higher quality demonstrations.

### B.4 COMPARISON WITH STAR-COT

In concurrent work closely related to ours, Zelikman et al. (2022) applies a similar methodology to ours in regenerating the reasoning chains for incorrect examples (we denote STaR-CoT in this paper). Although our work shares similar ideas to their approach, we differ in several key respects. First, their approach does not involve any contextual information from the previous step. Moreover, their approach provides the correct answer directly to the model, rather than using the correctness of the answer or error reasoning steps as bootstrapping information. It is more prone to generating erroneous reasoning chains despite arriving at the correct answer. Moreover, they continue fine-tuning the model using the corrected reasoning chains, which is impossible for LLMs with 175B parameters (such as ChatGPT (Ouyang et al., 2022)). We conduct inference on one arithmetic reasoning dataset GSM8K utilizing both Iter-CoT's first stage and STaR-CoT with the same questions shown in Table 5. We use the same LLMs and temperature to generate reasoning chains and answers. It is observed that although STaR-CoT generates the correct answer, the rationales are wrong, leading to confusion of the entire reasoning chain and reducing the overall performance (**80.9**% compared to **60.5**%).

### B.5 AVERAGE LENGTH OF REASONING CHAINS

We compare the average length of reasoning chains generated by different methods, which are demonstrated in Figure 10. The average length of the reasoning chains demonstrated by Iter-CoT is significantly higher than other methods on the three datasets (GSM8K, CSQA and Last Letter Concatation). These results provide solid evidence that the reasoning chains demonstrated by Iter-CoT are more comprehensive than those by other alternative methods.

| Final CoT | CoT |
|---|---|
| Q: Four classmates were comparing their ages based on their birth month. They found out that Jolyn is 2 months older than Therese while Therese is 5 months older than Aivo. Then, Leon is 2 months older than Aivo. How much older in months is Jolyn than Leon? (GSM8K) | |
| A: Reasoning process: We are given that Jolyn is 2 months older than Therese and Therese is 5 months older than Aivo. Let the ages of Therese and Aivo be T and A, respectively. We can write the ages of Jolyn, Therese, and Aivo as J = T+2, T = A+5, and A = A. Leon is 2 months older than Aivo, so L = A+2. We want to find the difference between Jolyn's age and Leon's age, so J - L = (T+2) - (A+2). Substituting for T and A using the equations above gives us (A+7) - (A+2) = 5. Therefore, Jolyn is 5 months older than Leon. Final answer: 5. ✔ | A: Let's think step by step. First let's say the ages of Therese and Aivo is T and A. T=A+5 J=T+2 L=A+2 J-L=T+2-(A+2) J-L=2 The correct answer is: 2. ✗ |
| Q: Take the last letters of the words in "Steve Sweeney" and concatenate them. (Letter(4)) | |
| A: Reasoning process: First, we need to split the sentence "Steve Sweeney" into individual words by looking for the spaces. This gives two words: "Steve" and "Sweeney". Next, we need to take last letter of each word, which are "e" and "y". Finally, we need to concatenate these letters together to get the final answer. Final answer: ey. ✔ | A: Let's think step by step. First we take the last letters of the words in "Steve Sweeney", which are "ete eeney". Since we need to concatenate them, we will end up with: "ete eeney" Therefore, the correct answer is: "ete eeney". ✗ |
| Q: May 6, 1992 is like yesterday to Jane, but that is actually ten years ago. What is the date one week from today in MM/DD/YYYY? Choices: A.05/13/2044 B.06/10/2002 C.05/13/2002 D.03/13/2003 E.11/13/2001 (Date Understanding) | |
| A: Reasoning process: 1. May 6, 1992 is ten years ago from today, which means today is May 6, 2002. 2. The date one week from today is the date one week from May 6, 2002. 3. The date one week from May 6, 2002 is May 13, 2002. 4. The date one week from today in MM/DD/YYYY is 05/13/2002. Final answer: C.05/13/2002. ✔ | A: Let's think step by step. The date one week from today is the date one week from Monday May 13, 2002. Thus the correct answer is: B.06/10/2002. ✗ |

Table 4: This is an example of Iter-CoT compared to CoT on three different reasoning datasets.

## C EXPERIMENT DETAILS

### C.1 DATASETS AND TASKS

We evaluate Iter-CoT using ten datasets from three different categories of reasoning tasks. The specific descriptions, divisions, and references of each dataset are shown in Table 6. The examples of each reasoning task are shown in Table 7.

## D FULL EXEMPLARS GENERATED BY ITER-COT

| Final CoT | STaR CoT |
|---|---|
| Q: Tina makes $18.00 an hour. If she works more than 8 hours per shift, she is eligible for overtime, which is paid by your hourly wage + 1/2 your hourly wage. If she works 10 hours every day for 5 days, how much money does she make? (GSM8K) | |
| A: Reasoning process: First, let's figure out how many hours Tina works per week. She works 10 hours per day and 5 days per week, so 10*5 = 50 hours. Next, let's figure out how many of those hours are overtime. She is eligible for overtime if she works more than 8 hours per shift and she works 10 hours per shift, so 10-8 = 2 hours of overtime per shift. And since Tina works 5 shifts and 2 hours of overtime per shift, she works a total of 5*2 = 10 hours of overtime. Now we can figure out how much Tina makes in regular pay. She works 50-10 = 40 hours per week at her regular wage of $18 per hour, so she makes 40*18 = $720 in regular pay. Now we can figure out how much Tina makes in overtime pay. She gets time and a half for overtime, so 1.5 * $18 per hour = $27 per hour. And since she works 10 hours of overtime and makes $27 per hour, she makes 10*27 = $270 in overtime pay. So in total, Tina makes 720 + 270 = $990 per week. ✔ | A: The correct answer is: 990. Let's think step by step. Tina works 10 hours per day for 5 days, so she works 10*5 = 50 hours. She is eligible for overtime after 8 hours per shift, so she works 50-8 = «50-8=42»42 hours of overtime. She makes $18.00 per hour, so she makes 18*42 = 756 dollars of overtime pay. Overtime pay is her hourly wage + 1/2 her hourly wage, so she makes 18+(18/2) = $27.00 per hour of overtime. She works 42 hours of overtime, so she makes 27*42 = 1134 dollars of overtime pay. She makes $18.00 per hour for her regular hours, so she makes 18*8 = «18*8=144»144 dollars of regular pay. She makes a total of 144+1134 = 1278 dollars. ✗ Therefore, the correct answer is: 990. ✔ |

Table 5: This is an example of Final CoT compared to STaR-CoT on GSM8K. The result shows that STaR-CoT is prone to generate erroneous reasoning chains despite arriving at the correct answer.

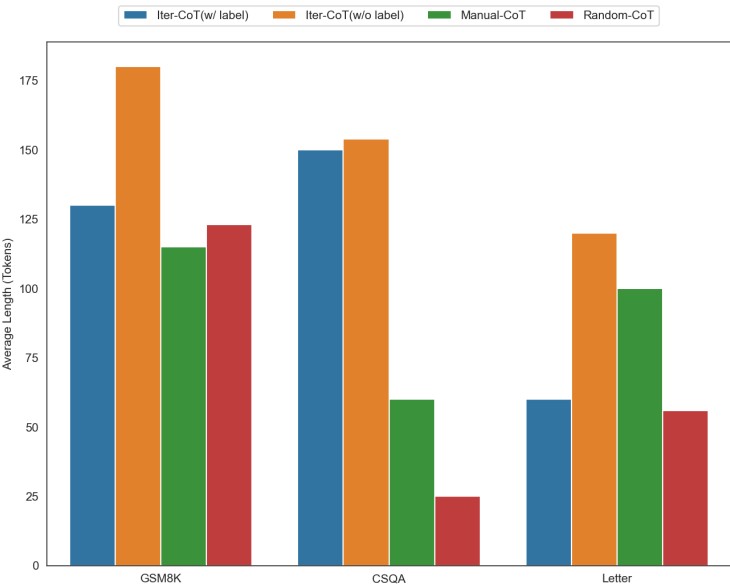

Figure 10: Average length of reasoning chains generated by different methods on GSM8K, CSQA and Last Letter Concatenation.

| Dataset | Answer Format | Examples | Train | Test | Licence |
|---------|---------------|----------|-------|------|---------|
| GSM8K | Number | 8 | 7473 | 1319 | MIT License |
| AQuA | Multiple choice | 4 | 97467 | 254 | Apache-2.0 |
| AddSub | Number | 8 | GSM8K* | 395 | - |
| SingleEq | Number | 8 | GSM8K* | 508 | - |
| SVAMP | Number | 8 | GSM8K* | 1000 | MIT License |
| ASDiv | Number | 8 | GSM8K* | 2096 | - |
| CSQA | Multiple choice | 7 | 9741 | 1221 | - |
| StrategyQA | Bineary | 6 | 2821 | 1880 | Apache-2.0 |
| Date | Multiple choice | 8 | 69 | 300 | Apache-2.0 |
| Letter(4) | String | 4 | 500(2*) | 500(4*) | Apache-2.0 |

Table 6: The statistics of the datasets used in this paper. Examples are the number of examples demonstrations for each dataset. GSM8K* denotes constructed the training set using the GSM8K, cause no available training set for the current dataset. 2* and 4* in the "Letter(4)" row refers to using 2 letters in the training set while using 4 letters in the test set (Out-of-Domain).

| Reasoning Tasks | Example |
|-----------------|---------|
| Arithmetic | Marissa is hiking a 12-mile trail. She took 1 hour to walk the first 4 miles, then another hour to walk the next two miles. If she wants her average speed to be 4 miles per hour, what speed (in miles per hour) does she need to walk the remaining distance? |
| Commonsense | Sammy wanted to go to where the people were. Where might he go? Choices: A.race track B.populated areas C.the desert D.apartment E.roadblock |
| Symbolic | Take the last letters of each words in "Geo Kody Isaias Giovanni" and concatenate them. |

Table 7: The examples are sampled from the test sets of three typical reasoning datasets, GSM8K (Cobbe et al., 2021), CommonSenseQA (Talmor et al., 2019) and Last Letter Concatation (Wei et al., 2022), respectively.

| DATASET | Iter-CoT Exemplars |
|---------|---------------------|
| GSM8K | Q: Sheila, Purity, and Rose want to rent a house. Sheila has offered to pay five times Purity's share of the rent. Rose can only afford thrice what Purity pays. If Rose's share is $1,800, what is the total house rent?
A: Reasoning Process: We are given that Rose's share of the rent is $1800, and that Rose can only afford thrice what Purity pays. Therefore, Purity's share of the rent is 1800 / 3 = $600. We are also given that Sheila has offered to pay five times Purity's share of the rent. Therefore, Sheila's share is 5 * 600 = $3000. Finally, we have Rose's share of $1800, Purity's share of $600 and Sheila's share of $3000. Therefore, the total rent is 1800 + 600 + 3000 = $5400. Final answer: 5400.

Q: Padma is trading cards with Robert. Padma started with 75 cards and traded 2 or her valuable ones for 10 of Robert's cards. Robert started with 88 of his own cards and traded another 8 of his cards for 15 of Padma's cards. How many cards were traded between both Padma and Robert?
A: Reasoning Process: Padma started with 75 cards. She traded 2 cards for 10 cards, so she now has 75-2+10=83 cards.
Robert started with 88 cards. He traded 8 cards for 15 cards, so he now has 88-8+15=95 cards.
The total number of cards traded between both Padma and Robert is 2+8+10+15=35 cards.
This includes the 2 cards that Padma traded for 10 cards, the 8 cards that Robert traded for 15 cards, and the 10 and 15 cards that were traded between them. Final answer: 35.

Q: Juanita goes through 1 bottle of sunscreen a month. She likes to stock up for the entire year when a certain website offers 30% off her entire purchase. If each bottle is $30.00 how much will all of the sunscreen cost after the discount?
A: Reasoning Process: Juanita needs 1 bottle of sunscreen a month, and there are 12 months in a year, so she needs 12 bottles for the year. The website is offering a 30% discount on her entire purchase. Each bottle of sunscreen costs $30.00. To find the total cost of the sunscreen after the discount, we first need to find the discount amount. This is done by multiplying the price per bottle by the discount rate and the number of bottles: $30.00 * 0.3 * 12 = $108.00. Next, we need to find the price before the discount. This is done by multiplying the price per bottle by the number of bottles: $30.00 * 12 = $360.00. Finally, we subtract the discount amount from the total price before the discount to find the final price after the discount: $360.00 - $108.00 = $252.00. Final answer: 252.

Q: On a three-day trip, Wendy drove 125 miles on the first day, and 223 miles on the second day. How many miles did she drive on the third day, if the total miles that Wendy drove for the trip is 493 miles?
A: Reasoning Process: 1. Wendy drove 125 miles on the first day and 223 miles on the second day, so the total miles she drove on the first two days is 125+223 = 348 miles.
2. The total miles Wendy drove on the trip is 493 miles.
3. Therefore, the remaining miles she drove on the third day is 493-348 = 145 miles. Final answer: 145.

Q: Mrs. Lopez and her family are going to the movie theatre. Adult tickets are $11. Children's tickets (ages 3-12) are $8. Senior citizen's tickets (ages 60+) are $9. Mrs. Lopez needs to buy movie tickets for her husband, herself, her parents (ages 72 and 75), and her three children (ages 7, 10, 14). How much money does she need to buy everyone's tickets?
A: Reasoning Process: 1. Mrs. Lopez needs to buy tickets for her husband and herself. The cost for these tickets will be $11+$11=$22.
2. Mrs. Lopez needs to buy tickets for her parents, who are senior citizens. The cost for these tickets will be $9+$9=$18.
3. Mrs. Lopez needs to buy tickets for her three children, who are between the ages of 3-12. The cost for these tickets will be $8+$8+$8=$24.
The total cost of all the tickets will be $22+$18+$24=$64. Final answer: 64.

Q: Mark has 30 candies, Peter has 25 candies, and John has 35 candies. They decided to combine their candies together and share them equally. How many candies will each one of them have?
A: Reasoning Process: 1. Mark has 30 candies, Peter has 25 candies, and John has 35 candies.
2. We need to add the number of candies that each person has to find the total number of candies they have.
3. So, 30 + 25 + 35 = 90 candies.
4. Then, we need to divide the total number of candies by the number of people, which is 3. 5. So, 90 / 3 = 30 candies. Final answer: 30. |

Table 8: The exemplars are selected on GSM8K train set. They are transferred to AddSub, SingleEq, SVAMP and ASDiv.

| DATASET | Iter-CoT Exemplars |
|---|---|
| GSM8K | Q: Ben's potato gun can launch a potato 6 football fields. If a football field is 200 yards long and Ben's dog can run 400 feet/minute, how many minutes will it take his dog to fetch a potato he launches? 
 A: Reasoning Process: To find the total distance that the potato can travel, we need to multiply the number of football fields by the number of feet per football field. Since there are 3 feet per yard, and there are 200 yards per football field, we have 200*3 = 600 feet per football field. Therefore, the total distance that the potato can travel is 6*600 = 3600 feet. Since Ben's dog can run at a rate of 400 feet per minute, it will take the dog 3600/400 = 9 minutes to fetch the potato. Final answer: 9. 

 Q: Brittany has 3 gerbils: Puffy, Muffy, and Scruffy. Puffy weighs 5 ounces more than Muffy. Muffy weighs 3 ounces less than Scruffy. If Scruffy weighs 12 ounces, how much would the scale indicate, in ounces, if Brittany put Puffy and Muffy on the scale? 
 A: Reasoning Process: We are given that Scruffy weighs 12 ounces and that Muffy weighs 3 ounces less than Scruffy. We are also told that Puffy weighs 5 ounces more than Muffy. 
 Using this information, we can calculate the weight of Muffy by subtracting 3 ounces from Scruffy's weight of 12 ounces, giving us a weight of 12-3 = 9 ounces for Muffy. 
 We can then calculate the weight of Puffy by adding 5 ounces to Muffy's weight of 9 ounces, giving us a weight of 9+5 = 14 ounces for Puffy. 
 If Brittany puts Puffy and Muffy on the scale, the scale would indicate a weight of 14+9 = 23 ounces. Final answer: 23. |

Table 9: (Cont.) The exemplars are selected on GSM8K train set. They are transferred to AddSub, SingleEq, SVAMP and ASDiv.

| DATASET | Iter-CoT Exemplars |
|---|---|
| AQuA | Q: What number has a 5:1 ratio to the number 11? Options: A:22 B:50 C:55 D:52 E:12 
 A: Reasoning Process: In order to find the number that has a 5:1 ratio to 11, we first need to understand what a ratio means. A ratio is a comparison of two numbers, and it is usually written in the form of a:b. In this case, the ratio is 5:1, which means that for every 5 units of the first number, there is 1 unit of the second number. Therefore, if we want to find the number that has a 5:1 ratio to 11, we need to find a number that is 5 times larger than 11. Then, we can check each of the given options to see if any of them are 5 times larger than 11. The only option that fits this description is C:55. Final answer: C. 

 Q: The average of seven numbers is 18. The average of first three numbers is 14 and the average of last three numbers is 23. What is the middle number? Options: A:25 B:27 C:15 D:32 E:34 
 A: Reasoning Process: 1. The average of the seven numbers is 18, so the sum of the seven numbers is $18 * 7 = 126$ 
 2. The average of the first three numbers is 14, so the sum of the first three numbers is $14 * 3 = 42$ 
 3. The average of the last three numbers is 23, so the sum of the last three numbers is $23 * 3 = 69$ 
 4. The sum of the first three numbers and the last three numbers is $42 + 69 = 111$ 
 5. The sum of the middle number and the sum of the first three numbers plus the last three numbers is equal to the sum of the seven numbers, in other words, $111 + x = 126$. 
 6. The middle number is $x = 15$ Final answer: C. 

 Q: Average score for Virat Kohli in a series of 10 matches is 38.9 runs. If the average for first six matches comes out to be 42 what is his average in the last 4 matches of the series? Options: A:34.25 B:34.28 C:24.252 D:64.28 E:34.21 
 A: Reasoning Process: 1) To find the average score for Kohli in the last 4 matches, we need to find the total number of runs he scored in these matches. 2) We know that his average score in the series of 10 matches is 38.9 runs. Therefore, he scored a total of 38.9 * 10 = 389 runs in these matches. 3) We are given that his average score in the first 6 matches is 42 runs. Therefore, he scored a total of 42 * 6 = 252 runs in these matches. 4) The total number of runs he scored in the last 4 matches is 389 - 252 = 137 runs. 5) Therefore, his average score in the last 4 matches is 137 / 4 = 34.25 runs. Final answer: A. 

 Q: A train runs at the speed of 72 kmph and crosses a 250 metre long platform in 26 seconds. What is the length of the train? Options: A:220 metre B:230 metre C:250 metre D:260 metre E:270 metre 
 A: Reasoning Process: We know that the train crosses a 250 metre long platform in 26 seconds at the speed of 72 kmph. To find the length of the train, we need to first convert the speed from kmph to m/s, so we use the formula: $V = V_{kmph} \times \frac{1}{3.6}$. Plugging in the values, we get: $V = 72 \times \frac{1}{3.6} = 20$ m/s. Now that we have the speed in m/s, we can use the formula: $D = Vt$ to find the distance the train covers in 26 seconds. Plugging in the values, we get: $D = 20 \times 26 = 520$ m. Since the length of the platform is 250 metre, and the train covers 520 metre in 26 seconds, the length of the train must be: $520 - 250 = 270$ m. Final answer: E. |

Table 10: The exemplars are selected on AQuA train set.

| DATASET | Iter-CoT Exemplars |
|---------|--------------------|
| CSQA | Q: Where could a fungus grow and not be disturbed by sunlight? Choices: A.under rocks B.manhattan C.toenails D.grocery store E.fallen tree
A: Reasoning process: 1. Fungi need moisture and shade to grow. 2. Rocks can provide both moisture and shade. 3. Therefore, rocks can be a good place for fungi to grow without being disturbed by sunlight. Final answer: A.

Q: The man wanted to take a bus, but all the chars were full of people doing what? Choices: A.sitting down B.refusing C.saving money D.sit down E.go outdoors
A: Reasoning process: 1. Eliminate answer C ("saving money") because it doesn't have anything to do with taking a bus. 2. Eliminate answer E ("go outdoors") because it doesn't make sense for people to be on a bus if they're already outside. 3. Choose answer A ("sitting down") because it's the only remaining answer that makes sense given the context clues in the question. Final answer: A.

Q: A beaver can destroy a machines functionality if they build their dam in this? Choices: A.strip club B.pocket C.millpond D.ontario E.lake or river
A: Reasoning process: 1. A beaver's dam is built in a body of water. 2. A millpond is a man-made pond used for a watermill. 3. Therefore, if a beaver builds its dam in a millpond, it can potentially destroy the functionality of the watermill. Final answer: C.

Q: How might a person address someone who is leaving? Choices: A.further cause B.wave goodbye C.attempt suicide D.insulting them E.thank god
A: Reasoning process: 1. We know that the person is leaving, so we can eliminate answer choices A and C because they don't make sense in context. 2. We can further eliminate answer choice D because it's rude to insult someone who is leaving. 3. That leaves us with answer choices B and E. 4. Out of these two, the more appropriate option would be B because it's a common way to show polite acknowledgement of someone leaving. Final answer: B.

Q: What is a characteristic of thin glass? Choices: A.break easily B.shattering C.melt D.bend E.hold water
A: Reasoning process: A: Break easily - This fits the characteristic of thin glass, as it is known for its fragility and tendency to break under pressure. B: Shattering - This could be a possible characteristic of thin glass, but it is not mentioned in the given choices, so we cannot select it as the answer. C: Melt - This does not fit the characteristic of thin glass, as it does not need to be heated in order to bend or shape it. D: Bend - This does not fit the characteristic of thin glass, as it is too fragile to be bent or shaped without breaking. E: Hold water - This does not fit the characteristic of thin glass, as it is not a property typically associated with glass. Final answer: A.

Q: What might be the result of a season of successful skiing? Choices: A.finish line B.broken bones C.broken legs D.chapped lips E.healthy body
A: Reasoning process: 1. A season of successful skiing could result in a healthy body if the person is exercising regularly and taking good care of their physical health while skiing. 2. None of the other options, such as a finish line, broken bones, or chapped lips, are related to the potential health benefits of skiing, so they cannot be the correct answer. 3. Therefore, the correct answer is E. healthy body. Final answer: E.

Q: Glass that hasn't been treated to be extra strong is what? Choices: A.weak B.fragile C.forceless D.regular E.flimsy
A: Reasoning process: 1. The question asks about glass that hasn't been treated to be extra strong. This means that the glass has not undergone any special processes or treatments to make it stronger than usual. 2. The answer choices are: A. weak, B. fragile, C. forceless, D. regular, and E. flimsy. 3. We need to find the answer choice that best describes glass that hasn't been treated to be extra strong. 4. "A. weak" means not strong. "B. fragile" means easily broken. "C. forceless" means without strength or power. "D. regular" means normal or usual. "E. flimsy" means weak and easily broken. 5. Based on these definitions, the correct answer is "B. fragile". Glass that hasn't been treated to be extra strong is likely to be fragile, or easily broken. This is because it has not undergone any special processes to make it stronger, so it will be more susceptible to breakage and damage. Other words that may be used to describe this type of glass include "delicate" or "brittle". Final answer: B. |

Table 11: The exemplars are selected on CSQA train set.

| DATASET | Iter-CoT Exemplars |
|---|---|
| StrategyQA | Q: Does a Toyota Prius require an anchor in order to park?
A: Reasoning process: A Toyota Prius is a car, and any car needs to be anchored to the ground in order to park, unless it is parked on a slope. Final answer: Yes.

Q: Did the husband of Elizabeth Hamilton work with a lieutenant?
A: Reasoning process: 1. First, we need to know who Elizabeth Hamilton is. Elizabeth Hamilton was the wife of Alexander Hamilton.
2. Then, we need to determine if Alexander Hamilton worked with a lieutenant. According to historical records, Alexander Hamilton did work with a lieutenant named John Laurens during the American Revolutionary War.
3. Therefore, the answer to the question "Did the husband of Elizabeth Hamilton work with a lieutenant?" is Yes. Final answer: Yes.

Q: Would Constantinople survive a hypothetical invasion by Charlemagne?
A: Reasoning process: 1. Constantinople was the capital of the Byzantine Empire, and Charlemagne was the ruler of the Carolingian Empire, which was in Europe.
2. Constantinople was a well-fortified city with strong walls and defenses.
3. Charlemagne was known for his military prowess and conquest.
4. The Byzantine Empire was a major power in its time, and Constantinople was a key strategic city for the empire.
5. However, the Carolingian Empire was also a major power and Charlemagne was a skilled and experienced military leader.
6. In a hypothetical situation, Constantinople may put up a strong resistance against Charlemagne's forces, but it would not be able to withstand a sustained attack. Final answer: Yes.

Q: Would King Henry V hypothetically prevail against Charlemagne at Agincourt?
A: Reasoning process: 1. First, we need to understand the question and identify the key information: "Did John Lennon collaborate with the filmmaker behind Walden?"
2. Next, we need to research what Walden is and who the filmmaker behind it is: Walden is a documentary film by Alan Watts.
3. With this information, we can now search for evidence of John Lennon collaborating with Alan Watts.
4. After conducting research, it is clear that John Lennon did collaborate with Alan Watts on a film project called 'The Touch of Zen'. Final answer: Yes.

Q: Did John Lennon collaborate with the filmmaker behind Walden?
A: Reasoning process: Monks are members of religious orders that usually take vows of poverty, chastity, and obedience. These vows are typically associated with a life of spiritual contemplation and non-violent pursuits. However, there are examples of monks engaging in warfare, such as the Knights Templar during the Crusades and Buddhist Shaolin monks who have defended temples during conquests. These examples suggest that it is not always the case that monks are forbidden from engaging in warfare. Final answer: Yes.

Q: Could Saint Augustine have read Hamlet?
A: Reasoning process: Saint Augustine was an early Christian saint who lived about 1,600 years ago.
Hamlet is a play by Shakespeare which is believed to have been written in the late 16th century.
It is possible that the play was written earlier than that and performed for Saint Augustine. Final answer: Yes. |

Table 12: The exemplars are selected on StrategyQA train set.

| DATASET | Iter-CoT Exemplars |
|---------|-------------------|
| Date Understading | Q: Yesterday was 12/31/1929. Today could not be 12/32/1929 because December has only 31 days. What is the date yesterday in MM/DD/YYYY? Choices: A.12/31/1929 B.12/31/1856 C.01/10/1930 D.04/01/1930 E.12/31/1838 F.01/03/1930
A: Reasoning process: 1. We know that yesterday was 12/31/1929.
2. Today cannot be 12/32/1929 because December only has 31 days.
3. Therefore, the date yesterday must be 12/31/1929. Final answer: A.

Q: The deadline is Jun 1, 2021, which is 2 days away from now. What is the date one week from today in MM/DD/YYYY? Choices: A.12/06/2021 B.05/30/2021 C.03/07/2021 D.06/05/2021 E.06/06/2021
A: Reasoning process: 1. The deadline is Jun 1, 2021, which is 2 days away from now. 2. Jun 1, 2021 is the date two days from now. 3. Adding 7 days from this date will give us Jun 6, 2021, which is the date one week from today. 4. Converting this date into MM/DD/YYYY format, we get 06/06/2021. Final answer: E.

Q: 2015 is coming in 36 hours. What is the date today in MM/DD/YYYY? Choices: A.12/08/2014 B.12/22/2014 C.12/26/2014 D.12/29/1974 E.12/29/2014 F.12/27/2014
A: Reasoning process: When we are given that 2015 is coming in 36 hours, we first need to determine what the current date is. We know that 36 hours is equal to 1.5 days, so we need to subtract 1.5 days from the date when 2015 arrives, which is 12/31/2014. This means that the current date is 12/29/2014. Final answer: E.

Q: May 6, 1992 is like yesterday to Jane, but that is actually ten years ago. What is the date one week from today in MM/DD/YYYY? Choices: A.05/13/2044 B.06/10/2002 C.05/13/2002 D.03/13/2003 E.11/13/2001
A: Reasoning process: 1. May 6, 1992 is ten years ago from today, which means today is May 6, 2002. 2. The date one week from today is the date one week from May 6, 2002. 3. The date one week from May 6, 2002 is May 13, 2002. 4. The date one week from today in MM/DD/YYYY is 05/13/2002. Final answer: C.

Q: The current local time is 3:02 pm of 5/4/2004. What is the date tomorrow in MM/DD/YYYY? Choices: A.05/05/1915 B.05/06/2004 C.01/05/2005 D.05/04/2004 E.02/15/2004 F.05/05/2004
A: Reasoning process: The current date is 5/4/2004, and tomorrow is the next day, which is 5/5/2004. This is because the current month is May and the current day is the 4th, so the next day is the 5th. The year remains the same because we are only looking at the date for tomorrow and not a longer time period. Final answer: F.

Q: Today is Sep 9, 1909. What is the date today in MM/DD/YYYY? Choices: A.09/09/1939 B.12/11/1909 C.09/09/1909 D.09/30/1909 E.11/19/1909 F.09/09/1886
A: Reasoning process: * The question is asking for the date today in MM/DD/YYYY format. * We know that today's date is September 9, which is 09. * The next step is to find the correct day. September has 30 days, so the date today is 09/30/1909. * However, this is not correct because the year is 1909, not 1939. * Therefore, the correct date today is 09/09/1909. Final answer: C.

Q: Today is the palindrome day of 2020, because the MMDDYYYY format of the date is the same backwards as forwards. What is the date 24 hours later in MM/DD/YYYY? Choices: A.02/03/1982 B.02/03/2100 C.02/03/2020 D.02/04/2020 E.02/03/2094 F.01/02/2020
A: Reasoning process: The palindrome date is of the form MM/DD/YYYY, and it is the same when read backwards. The date of today is 02/02/2020 and if we read it backwards it is still the same, so it is the palindrome date. The next day is the 02/03/2020 and the palindrome date of the day after tomorrow is the 02/04/2020. Final answer: C.

Q: The deadline is Jun 1, 2021, which is 2 days away from now. What is the date one week ago from today in MM/DD/YYYY? Choices: A.05/23/2054 B.05/24/2021 C.04/13/2021 D.05/23/1940 E.05/23/2021 F.05/23/2020
A: Reasoning process: If the deadline is Jun 1, 2021, which is 2 days away, then today is May 30, 2021. If today is May 30, 2021, then one week ago was May 23, 2021. Final answer: E. |

Table 13: The exemplars are selected on Date Understanding train set.

| DATASET | Iter-CoT Exemplars |
|---------|-------------------|
| Letter (4) | Q: Take the last letters of the words in "Agustin Lowe" and concatenate them.
A: Reasoning process: "Agustin Lowe" consists of two words, "Agustin" and "Lowe", and each of them has 5 and 4 letters, respectively. The last letters of these two words are "n" and "e", respectively. When we concatenate these two letters, we get "ne". Final answer: ne.

Q: Take the last letters of the words in "Vern Griffin" and concatenate them.
A: Reasoning process: 1. First, we need to take the last letters of each word in "Vern Griffin". 2. Then we need to concatenate the last letters to get the final answer. 3. The last letters of "Vern" and "Griffin" are "n" and "n". 4. Concatenating the last letters, we get "nn". Final answer: nn.

Q: Take the last letters of the words in "Randell Graham" and concatenate them.
A: Reasoning process: At the start of the match, Alice is playing goalkeeper, Bob is playing left winger, and Claire is playing right midfielder. 1. Take the last letters of the words in "Randell Graham" which would be "ll" and "m". 2. Concatenate them to form "lm". Final answer: lm.

Q: Take the last letters of the words in "Randal Holland" and concatenate them.
A: Reasoning process: 1. We need to find the last letters of the words in "Randal Holland" 2. "Randal" ends in "l" and "Holland" ends in "d" 3. The last letters of the words in "Randal Holland" are "ld" Final answer: ld. |

Table 14: The exemplars are selected on Last Letter Concatenation train set.

| DATASET | Iter-CoT(w/o label) Exemplars |
|---|---|
| GSM8K | Q: In a house, there are 16 cats. Two of them are white, and 25% of them are black. The rest of the cats are grey. How many grey cats are in this house?
A: Reasoning Process: Given: Total number of cats = 16 Number of white cats = 2 Percentage of black cats = 25% Step 1: Calculate the number of black cats. Number of black cats = (25/100) * 16 = 4 Step 2: Calculate the number of grey cats. Number of grey cats = Total number of cats - (Number of white cats + Number of black cats) Number of grey cats = 16 - (2 + 4) = 10 Final answer: 10.

Q: Ben starts each day with $50, spends $15 every day and saves the rest. After a few days, his mom doubled his total savings and his dad gave him an additional $10. If he now has $500, How many days have elapsed?
A: Reasoning Process: We know that Ben starts each day with $50 and spends $15 every day. So, his daily savings can be calculated as: Daily savings = $50 - $15 = $35 Letś denote the number of days that have elapsed as "d". Therefore, the total savings before his mom doubled it and his dad gave him an additional $10 can be calculated as: Total savings before mom doubled it and dad gave him an additional $10 = ($35 * d) + $10 According to the question, his total savings after his mom doubled it and his dad gave him an additional $10 is $500. So, we can write the equation: ($35 * d) * 2 + $10 = $500 Simplifying the equation, we get: ($70 * d) + $10 = $500 Subtracting $10 from both sides, we get: ($70 * d) = $490 Dividing both sides by $70, we get: d = $490 / $70 Calculating the value of d, we get: d = 7 Therefore, approximately 7 days have elapsed. Final answer: 7.

Q: There were 600 people in the stadium when the football game started. Before the game was over, one-fourth of the boys and one-eighth of the girls left early. How many people remained to see the end of the game if there were 240 girls at the beginning of the game?
A: Reasoning Process: To find out how many people remained to see the end of the game, we need to subtract the number of boys who left early and the number of girls who left early from the total number of people at the beginning of the game. Given: Total people at the beginning of the game: 600 Number of girls at the beginning of the game: 240 First, let's calculate the number of boys at the beginning of the game: Total people at the beginning of the game - Number of girls at the beginning of the game = Number of boys at the beginning of the game 600 - 240 = 360 Next, let's calculate how many boys left early: (1/4) * Number of boys at the beginning of the game = (1/4) * 360 = 90 boys Then, let's calculate how many girls left early: (1/8) * Number of girls at the beginning of the game = (1/8) * 240 = 30 girls Finally, to find out how many people remained to see the end of the game, subtract the number of boys who left early and the number of girls who left early from the total number of people at the beginning of the game: Total people at the beginning of the game - Number of boys who left early - Number of girls who left early = Number of people who remained at the end of the game 600 - 90 - 30 = 480 Final answer: 480.

Q: A basketball team played 40 games and won 70% of the games. It still had 10 games to play. How many games can they lose to win 60% of their games?
A: Reasoning Process: To solve the problem, we need to determine the maximum number of games the basketball team can lose to maintain a 60% win rate. Step 1: Calculate the total number of games played by adding the games already played (40) to the games remaining (10): Total number of games = 40 + 10 = 50 games Step 2: Calculate the number of games the team needs to win to achieve a 60% win rate: Number of games to win = 50 x 60% = 30 games Step 3: Calculate the number of games already won: Number of games already won = 40 x 70% = 28 games Step 4: Calculate the maximum number of games the team can lose: Maximum number of games that can be lost = Number of games to win - Number of games already won = 30 - 28 = 2 games. Final answer: 2.

Q: Jack will have ten times more handball trophies than Michael has right now in three years. If Michael has 30 trophies right now, and the number of his trophies increases by 100 in three years, what's the total number of trophies they'll have altogether after three years?
A: Reasoning Process: Given: - Michael currently has 30 handball trophies. - In three years, the number of Michael's trophies will increase by 100. To find the total number of trophies they'll have altogether after three years, we need to calculate the number of trophies Jack will have in three years and then add it to the number of trophies Michael will have in three years. Since Jack will have ten times more handball trophies than Michael in three years, we can calculate Jack's number of trophies as follows: Jack's trophies in three years = 10 * (Michael's trophies in three years) = 10 * (30 + 100) = 10 * 130 = 1300. Now, let's calculate the total number of trophies they'll have altogether after three years: Total trophies in three years = Michael's trophies in three years + Jack's trophies in three years = 130 + 1300 = 1430. Final answer: 1430.

Q: John decides to install a ramp in his house. He needs to get permits which cost $250. After that, he hires a contractor which costs $150 an hour and the guy works for 3 days at 5 hours per day. He also pays an inspector 80% less to make sure it is OK. How much was the total cost?
A: Reasoning Process: The cost of permits is $250. The contractor works for 3 days, with 5 hours per day, resulting in a total of 3 * 5 = 15 hours. The contractor charges $150 per hour, so the total cost of hiring the contractor is 15 * $150 = $2250. The inspector is paid 80% less, which means he is paid only 20% of the original cost. So, the cost of the inspector is 20/100 * $2250 = $450. To calculate the total cost, we need to add the cost of permits, contractor, and inspector: Total cost = Cost of permits + Cost of contractor + Cost of inspector = $250 + $2250 + $450 = $2950. Final answer: 2950. |

Table 15: The exemplars are selected on GSM8K train set. This set of exemplars is used by GSM8K, ASDiv, SVAMP, and SingleEq.

| DATASET | Iter-CoT(w/o label) Exemplars |
|---------|-------------------------------|
| GSM8K | Q: Ignatius owns 4 bicycles. A friend of his owns different types of cycles, which have three times are many tires as Ignatius's bikes have. He has one unicycle, a tricycle, and the rest are bikes. How many bicycles does the friend own? A: Reasoning Process: Ignatius owns 4 bicycles, and each bicycle has 2 tires. So Ignatius has a total of 4 bicycles * 2 tires per bicycle = 8 tires. The friend's cycles have three times as many tires as Ignatius's bikes. Therefore, the friend's cycles have 3 * 8 tires = 24 tires. We also know that the friend has one unicycle (1 tire) and one tricycle (3 tires). To find out how many bicycles the friend owns, we subtract the number of tires from the unicycle and tricycle from the total number of tires the friend's cycles have: 24 tires - 1 tire (unicycle) - 3 tires (tricycle) = 20 tires Since each bicycle has 2 tires, the number of bicycles the friend owns is 20 tires / 2 tires per bicycle = 10 bicycles. Therefore, the friend owns 10 bicycles. Final answer: 10. 

 Q: James spends 3 hours vacuuming and 3 times as long on the rest of his chores. How long does he spend on his chores? A: Reasoning Process: To find out how long James spends on his chores, we need to add the time he spends vacuuming (3 hours) to the time he spends on the rest of his chores. The rest of his chores take 3 times as long as vacuuming, so we can calculate that by multiplying the vacuuming time (3 hours) by 3. 3 hours (vacuuming) x 3 = 9 hours (rest of chores) Now, we add the vacuuming time and the rest of the chores time together: 3 hours (vacuuming) + 9 hours (rest of chores) = 12 hours Therefore, James spends a total of 12 hours on his chores. Final answer: 12. |

Table 16: (Cont.) The exemplars are selected on GSM8K train set. This set of exemplars is used by GSM8K, ASDiv, SVAMP, and SingleEq.

| DATASET | Iter-CoT(w/o label) Exemplars |
|---------|-------------------------------|
| AQuA | Q: There are a lot of houses such that the numbers of their doorplates are odd numbers and the first number of the doorplates is 545, the last number of the doorplates is 805. How many houses are there? Options: A)125 B)111 C)101 D)121 E)131 
 A: Reasoning process: To find the count of houses with odd doorplate numbers between 545 and 805, we need to find the difference between the two numbers and divide it by 2 (since we are incrementing by 2 for odd numbers) and then add 1 to include the first number. The difference between 805 and 545 is 260. Dividing 260 by 2 gives us 130. Adding 1 to include the first number gives us a total of 131 houses. Final answer: E. 

 Q: The ratio of the radius of two circles is 2: 3, and then the ratio of their areas is? Options: A)1: 8 B)4:9 C)1: 9 D)1: 3 E)1: 2 
 A: Reasoning process: To find the ratio of the areas of two circles, we need to square the ratio of their radii. Given that the ratio of the radii is 2:3, the ratio of their areas would be $(2\hat{2}):(3\hat{2})$, which simplifies to 4:9. Final answer: B. 

 Q: A train passes a station platform in 36 seconds and a man standing on the platform in 20 seconds. If the speed of the train is 54 km/hr, what is the length of the platform? Options: A)388 B)240 C)88 D)66 E)221 
 A: Reasoning process: Given: - Speed of the train = 54 km/hr - Train passes a station platform in 36 seconds - Train passes a man standing on the platform in 20 seconds Let's denote: - Length of the train as 'L' - Length of the platform as 'P' When the train passes the platform, it covers a distance equal to the length of the train plus the length of the platform. Distance covered when passing the platform = L + P The train covers this distance in 36 seconds. Speed = Distance / Time Therefore, (L + P) / 36 = 54 km/hr To convert km/hr to m/s, we multiply by (5/18): (L + P) / 36 = 54 * (5/18) m/s Simplifying, we have: (L + P) / 36 = 15 m/s Similarly, when the train passes the man, it covers a distance equal to the length of the train. Distance covered when passing the man = L The train covers this distance in 20 seconds. Speed = Distance / Time Therefore, L / 20 = 54 * (5/18) m/s Simplifying, we have: L / 20 = 15 m/s Now, we have two equations: 1) (L + P) / 36 = 15 2) L / 20 = 15 From equation 2, we can express L in terms of P: L = 20 * 15 = 300 Substituting this value of L in equation 1: (300 + P) / 36 = 15 Simplifying, we have: 300 + P = 15 * 36 300 + P = 540 P = 540 - 300 P = 240 Therefore, the length of the platform is 240 meters. Final answer: B. 

 Q: A fruit seller had some apples. He sells 40% apples and still has 420 apples. Originally, he had Options: A)588 apples B)600 apples C)672 apples D)700 apples E)None 
 A: Reasoning process: The fruit seller sells 40% of his apples and still has 420 apples remaining. Let's denote the original number of apples as x. After selling 40% of x, the remaining number of apples is 0.6x. According to the given information, 0.6x = 420. To find the value of x, we divide both sides of the equation by 0.6: x = 420 / 0.6 = 700. Final answer: D. |

Table 17: The exemplars are selected on AQuA train set.

| DATASET | Iter-CoT(w/o label) Exemplars |
|---------|-------------------------------|
| CSQA | Q: The Coast Guard is a police force that often does what? Choices: A.board vessels B.direct traffic C.ocean D.force law E.arrest
A: Reasoning process: The Coast Guard is a branch of the military that is responsible for maritime law enforcement and search and rescue operations. They are often involved in activities related to the sea and vessels. Among the options provided, the most suitable answer is A. board vessels. The Coast Guard frequently boards vessels for inspections, investigations, and enforcement of maritime laws and regulations. Final answer: A.

Q: Where do I store folding chairs for religious experiences? Choices: A.closet B.church C.room D.basement E.garage
A: Reasoning process: To determine the best option for storing folding chairs for religious experiences, we need to consider the specific requirements and context of religious activities. Option A: Closet - This could be a viable choice if there is a dedicated closet within the religious establishment specifically for storing folding chairs. It would ensure that the chairs are easily accessible and organized. Option B: Church - While a church may have designated areas for storing chairs, such as a storage room or basement, it may not be the most suitable choice if the religious experiences are held in other locations outside of the church. Option C: Room - This option is quite generic and lacks specificity. It could potentially be a suitable choice if there is a dedicated room within the religious establishment specifically for storing folding chairs. Option D: Basement - Similar to option B, a basement could be a suitable choice if the religious establishment has a dedicated basement area for storing chairs. However, if the religious experiences are held in other locations, this may not be the most practical option. Option E: Garage - Storing folding chairs in a garage may not be ideal for religious experiences as it is typically used for parking vehicles and may not provide a suitable and organized space for the chairs. onsidering the various options, the most suitable choice for storing folding chairs for religious experiences would be A. closet. This option ensures that the chairs are easily accessible and organized within the religious establishment. Final answer: A.

Q: What might people do to protect their legs from getting dirty on the farm? Choices: A.wear jeans B.milk cow C.believe in god D.avoid mud E.plant flowers
A: Reasoning process: To protect their legs from getting dirty on the farm, people can take several measures. Let's analyze the options provided: A. wear jeans - This option seems like a practical choice as jeans are known for their durability and ability to provide protection against dirt and mud. B. milk cow - While milking a cow is a common farm activity, it does not directly address the question of protecting legs from getting dirty. C. believe in god - This option does not appear to be relevant to protecting legs from getting dirty on the farm. D. avoid mud - Avoiding mud can be a preventive measure, but it may not always be possible or practical on a farm. E. plant flowers - Planting flowers does not address the question of protecting legs from getting dirty. Final answer: A.

Q: What may jogging make you feel like? Choices: A.increased heart rate B.improved health C.frightened D.shin splints E.being tired
A: Reasoning process: When you go jogging, your heart rate typically increases as your body works harder to meet the demands of the exercise. So, option A (increased heart rate) seems like a plausible choice. Jogging is also known to have numerous health benefits, such as improving cardiovascular fitness, strengthening muscles, and aiding in weight loss. Therefore, option B (improved health) is also a likely option. However, jogging is not typically associated with feelings of being frightened or experiencing shin splints. Shin splints are a common overuse injury that can occur during running or high-impact activities, but they are not directly related to jogging itself. Lastly, jogging can often lead to feelings of tiredness, especially if you push yourself and exert a lot of energy during the activity. So, option E (being tired) is a reasonable choice as well. Final answer: E.

Q: The best way to cure boredom at a high elevation with no buildings around is to do what? Choices: A.do crossword puzzle B.go skiing C.play games D.hang out at bar E.sleep
A: Reasoning process: In a high elevation with no buildings around, the best way to cure boredom would be to go skiing. Skiing is a popular recreational activity that can be enjoyed in mountainous areas, especially at high elevations. It provides an exhilarating experience and allows you to make the most of the natural surroundings. Final answer: B.

Q: Where is an inexpensive place to have a reception? Choices: A.outside B.wedding party C.restaurant D.church basement E.country club
A: Reasoning process: Based on the given choices, the most inexpensive option for having a reception would likely be a church basement. Church basements are often available for a lower cost or even for free for members of the church or community. This makes them a budget-friendly option for hosting events such as wedding receptions. Final answer: D.

Q: Where does the red cross respond to a storm? Choices: A.summer B.tropics C.blue shield D.weather report E.america
A: Reasoning process: The correct answer is E. America. The Red Cross responds to storms in various locations, including the United States, where they provide assistance and support during natural disasters such as storms and hurricanes. Final answer: E. |

Table 18: The exemplars are selected on CommonSenseQA train set.

| DATASET | Iter-CoT(w/o label) Exemplars |
|---|---|
| STQA | Q: Is the Antarctic Peninsula south of the Equator?
A: Reasoning process: The Antarctic Peninsula is located at the southernmost part of the Earth, which means it is in the southern hemisphere. The Equator is an imaginary line that divides the Earth into the northern and southern hemispheres. Therefore, since the Antarctic Peninsula is located in the southern hemisphere, it is south of the Equator. Final answer: Yes.

Q: Is lunch on the beach a good activity to spot the full circle of a rainbow?
A: Reasoning process: First, to see a rainbow, you need to be outside in the rain or near a source of water. If you are on the beach, you are likely to be near a source of water, so that criterion is satisfied. Second, you need to have the sun shining. If you are having lunch on the beach, you are likely to be there during the daytime, when the sun is shining, so that criterion is also satisfied. Third, you need to have the right angle between the sun, your eyes, and the water droplets in the air. This is a bit harder to predict, but if the conditions are right, you should be able to see a rainbow. However, as mentioned in the previous answer, the full circle of a rainbow cannot usually be seen from ground level, and lunch on the beach occurs at midday when the sun is likely high in the sky. Therefore, it is unlikely that you would be able to see the full circle of a rainbow from the beach during lunch. Final answer: No.

Q: Were any of the characters in Macbeth played by an Oscar-winning actor?
A: Reasoning process: To answer the question, we need to research the actors who have played the characters in Macbeth and find out if any of them have won an Oscar. Some of the characters in Macbeth have been played by Oscar-winning actors. For instance, the character of Macbeth has been played by actors such as Laurence Olivier, Orson Welles, and Michael Fassbender, all of whom have won Oscars for their performances in other roles. Final answer: Yes.

Q: Are monks forbidden from engaging in warfare?
A: Reasoning process: Monks are members of religious orders that usually take vows of poverty, chastity, and obedience. These vows are typically associated with a life of spiritual contemplation and non-violent pursuits. However, there are examples of monks engaging in warfare, such as the Knights Templar during the Crusades and Buddhist Shaolin monks who have defended temples during conquests. These examples suggest that it is not always the case that monks are forbidden from engaging in warfare. Final answer: No.

Q: Does Post Malone have a fear of needles?
A: Reasoning process:We know that Post Malone is a person because the question states that he is a person. People can have fears because it is a common human trait to have fears of various things. The fear of needles is a common fear among people because needles can cause pain, discomfort, and anxiety for many people. However, the fact that Post Malone has many tattoos on his body suggests that he is comfortable with needles and therefore does not have a fear of needles. Additionally, the fact that the most common method of tattooing involves electric tattoo machines and multiple needles suggests that Post Malone has been exposed to needles many times and has not shown any fear or discomfort. Therefore, it is unlikely that Post Malone has a fear of needles. Final answer: No.

Q: Can you get negative effects from a placebo?
A: Reasoning process: Placebos are substances or treatments that have no therapeutic effect, and are often used in medical trials as a control group to compare the effects of a real treatment. Since placebos themselves have no active ingredients, it is unlikely for them to cause negative effects on their own. However, it is possible for a person to experience negative effects from a placebo due to the nocebo effect. This occurs when a person believes a treatment to be harmful, even though it has no active ingredients, and they experience negative side effects as a result of their expectations and beliefs about the treatment. It is possible for a person to experience negative effects from a placebo due to the nocebo effect. Final answer: Yes. |

Table 19: The exemplars are selected on StrategyQA train set.

| DATASET | Iter-CoT(w/o label) Exemplars |
|---|---|
| Date Understading | Q: This is the last day of 1899. What is the date one week ago from today in MM/DD/YYYY? Choices: A.12/24/1802 B.12/24/1899 C.12/17/1899 D.01/02/1900 E.03/12/1900 F.05/24/1900
A: Reasoning process: To find the date one week ago from today, we need to subtract 7 days from the last day of 1899, which is December 31, 1899. Subtracting 7 days from December 31, 1899, we get December 24, 1899. Final answer: B.

Q: Today is the last day of the first quarter of 2008. What is the date a month ago in MM/DD/YYYY? Choices: A.02/01/2008 B.12/26/2007 C.02/28/2008 D.02/29/2008 E.02/08/2008 F.08/29/2008
A: Reasoning process: To find the date one month ago from the last day of the first quarter of 2008, we need to subtract one month from March 31, 2008. Since February only has 29 days in a leap year, we can calculate the answer is 02/29/2008 Final answer: D.

Q: Jane and John married on Jan 2, 1958. Today is their golden wedding anniversary. What is the date yesterday in MM/DD/YYYY? Choices: A.02/03/2008 B.12/27/2007 C.01/02/2008 D.12/31/2007 E.01/01/2008
A: Reasoning process: Jane and John married on Jan 2, 1958. Today is their golden wedding anniversary, which means they have been married for 50 years. To find the date yesterday, we need to subtract one day from the current date. Given that today is Jan 2, 2008, the correct answer is: 01/01/2008. Final answer: E.

Q: Today is 3/5, and it is Jane's second time in the year 1973 to see a meteor shower. What is the date 24 hours later in MM/DD/YYYY? Choices: A.03/02/1973 B.03/06/1973 C.04/05/1973 D.01/02/1973 E.03/05/1972 F.03/08/1983
A: Reasoning process: Today is 3/5, and we want to find the date 24 hours later. Adding 24 hours to 3/5 would give us 3/6. Final answer: B.

Q: Jane thinks today is 6/18/2019, but John thinks today is 6/19/2019. Jane is correct. What is the date one year ago from today in MM/DD/YYYY? Choices: A.06/17/2018 B.06/19/2018 C.06/13/2018 D.05/23/2018 E.07/09/2018 F.06/18/2018
A: Reasoning process: To find the date one year ago from today, we subtract 1 year from the current date. Given that today is 6/18/2019, one year ago would be 6/18/2018. Final answer: F.

Q: Jane scheduled 3 apointments with 5 poeple for tomorrow (Tue, 7/9/1972). What is the date one year ago from today in MM/DD/YYYY? Choices: A.07/09/1971 B.07/15/1971 C.07/08/1910 D.07/01/1971 E.07/07/1971 F.07/08/1971
A: Reasoning process: To find the date one year ago from today, we need to subtract 1 year from the given date, which is July 9th, 1972. Final answer: A.

Q: In the US, Thanksgiving is on the fourth Thursday of November. Today is the US Thanksgiving of 2001. What is the date one week from today in MM/DD/YYYY? Choices: A.11/30/2002 B.11/29/2078 C.12/27/2002 D.12/11/2002 E.11/29/2002 F.11/15/2002
A: Reasoning process: The US Thanksgiving in 2001 was on November 22nd. Adding one week to that date, we get November 29th. Final answer: E.

Q: The current local time is 3:02 pm of 5/4/2004. What is the date tomorrow in MM/DD/YYYY? Choices: A.05/05/1915 B.05/06/2004 C.01/05/2005 D.05/04/2004 E.02/15/2004 F.05/05/2004
A: Reasoning process: The current local time is 3:02 pm of 5/4/2004. To find the date tomorrow, we need to add one day to the current date. Since it is already 3:02 pm, tomorrow's date will be the same as the current date until 3:02 pm. After 3:02 pm, it will be the next day. Final answer: F. |

Table 20: The exemplars are selected on Date Understanding train set.

| DATASET | Iter-CoT(w/o label) Exemplars |
|---|---|
| Letter(4) | Q: Take the last letters of the words in "Keith Ruiz" and concatenate them.
A: Reasoning process: 1) The last letters of "Keith Ruiz" are "h" and "z". 2) Concatenating these letters gives us "hz". Therefore, the correct answer is: hz. Final answer: hz.

Q: Take the last letters of the words in "Saul Moss" and concatenate them.
A: Reasoning process: Let's take the last letters of each word in "Saul Moss": - The last letter of "Saul" is "l". - The last letter of "Moss" is "s". Now, let's concatenate them: - "l" + "s" = "ls" Final answer: ls.

Q: Take the last letters of the words in "Morton Stanton" and concatenate them.
A: Reasoning process: Taking the last letters of each word in "Morton Stanton" and concatenating them, we have: n (from Morton) n (from Stanton) Final answer: nn.

Q: Take the last letters of the words in "Drew Barnett" and concatenate them.
A: Reasoning process: Taking the last letters of the words "Drew" and "Barnett", we have "w" and "t" respectively. Concatenating them gives us "wt". Final answer: wt. |

Table 21: The exemplars are selected on Last Letter Concatation(4) train set.

