# OpenReview forum: "Enhancing Chain-of-Thoughts Prompting with Iterative Bootstrapping in Large Language Models"
_ICLR.cc/2024/Conference — ICLR 2024 Conference Withdrawn Submission_

### Official Review · Reviewer_TzhJ · 2023-11-01

**Soundness:** 3 good
**Presentation:** 3 good
**Contribution:** 3 good
**Rating:** 6
**Confidence:** 4

**Summary:**

This paper proposes an iterative bootstrapping method called Iter-CoT in chain-of-thoughts prompting for large language model reasoning. The major innovation of Iter-CoT is to self-correct the errors in reasoning chains by leveraging iterative bootstrapping and obtaining more precise and comprehensive reasoning chains. Experimental results on ten reasoning datasets among three different reasoning tasks demonstrate that our approach significantly outperforms the previous methods, achieving new state-of-the-art results.

**Strengths:**

1. This paper proposes iter-CoT, which uses the experiences with wrong cases to construct the demonstrations for CoT prompts.

2. It is a new CoT method that focuses on the demonstration selection to improve the in-context learning performance.

3. The evaluation is conducted on a wide range of benchmarks and various LLMs, showing advances in performance.

**Weaknesses:**

1. The idea of this paper is not very exciting.

a) The idea of self-correction is discussed in recent studies [1,2]. By constructing the correction demonstration pool, the samples that the LLM is prone to getting wrong are gathered and a correct CoT is prepared for each. By sampling from these, the model can get stronger prompt.

b) The idea of ‘bootstrap’ is not very appealing. First, LLM bootstrap has been proposed in [3], where the demos are totally generated by the proposed system. However, in iter-CoT, both ‘w/ label’ setting and ‘wo label’ setting are under supervised, i.e., the golden label or another more powerful LLM.

2.  (Minor) The analysis does not make this paper more convincing. The settings of demo selection in the ablation can hardly uncover what the model learns from history errors and corrections. The ‘rising and then falling’ trend in Figure 6 is not obvious. Also, the comparisons are coarse-grained. The demos, method, and reasoning steps are all different across all settings and may require in-depth analysis.

[1] Learning from Mistakes via Interactive Study Assistant for Large Language Models

[2] Large Language Models Cannot Self-Correct Reasoning Yet

[3] Demonstrate-Search-Predict: Composing retrieval and language models for knowledge-intensive NLP

**Questions:**

n/a

---

> ### Author Response · Authors · 2023-11-17
>
> **Weaknesses1-A**:  The idea of self-correction is discussed in recent studies.
>
> **Answer**:  While both of the referenced works mentioned the self-correction or self-improvement capabilities of LLM, aligning with our conceptual framework, it is crucial to note that our research are contemporaneous to these two works (Moreover, the paper "Large Language Models Cannot Self-Correct Reasoning Yet" is after submission of ICLR). Furthermore, there is a key distinction between our work and the first work you mentioned: They collect samples where the LLM first made mistakes in the initial stage so that the assitant can avoid making the same mistakes when reasoning about new questions. However, our work focuses on generating accurate and high-quality samples. In the inference process, we sampled only once and maintained consistency in subsequent inferences.
>
>
> **Weaknesses1-B**:  The idea of ‘bootstrap’ is not very appealing.
>
> **Answer**:  While our approach incorporates labels or soft labels, it is merely intended to provide the model with information about the correctness of its answers, aiming to facilitate self-correction. Your feedback has alerted us that 'bootstrap' may not be the most suitable term in this context. We should consider using other, more appropriate terms such as 'guidance.' We sincerely appreciate your recommendation.
>
>
> **Weaknesses2**:  The analysis does not make this paper more convincing.
>
> **Answer**:  We conduct an "Init-wrong-CoT" setting, which uses the exemplars generated in the initialization step, and their questions are the same as those used in Correct-CoT and Iter-CoT, so their reasoning chains are wrong. The comparison between Init-Wrong-CoT and Correct-CoT shows that error correction has significantly improved the final performance (7.9%). At the same time, the comparison between Correct-CoT and Iter-CoT illustrates the importance of summary. (2.8%).
>
> | Method            | GSM8K | Date | Letter | **Avg.** |
> |-------------------|-------|------|--------|----------|
> | Init-wrong-CoT    | 71.9  | 61.2 | 72.1   | 68.4     |
> | Init-correct-CoT  | 72.5  | 63.0 | 78.6   | 71.4     |
> | Random-CoT        | 72.6  | 64.5 | 73.2   | 70.1     |
> | Correct-CoT       | 79.2  | 67.0 | 82.6   | 76.3     |
> | Best-of-N-CoT     | 76.3  | 66.7 | 77.4   | 73.5     |
> | Iter-CoT          | 80.9  | 71.3 | 85.2   | 79.1     |
>
> In the above table, analysis yields the following conclusions:
>
> 1. Incorrect examples adversely impact model reasoning: Comparing Init-wrong-CoT (all examples incorrect), Random-CoT (some examples may be incorrect), and Init-correct-CoT (all examples correct), the results are 68.4% -> 70.1% -> 71.4% (varied questions in examples used in the above settings). Contrasting Init-wrong-CoT with Correct-CoT, the results are 68.4% -> 76.3% (same questions in examples used in the above settings).
>
> 2. The correction phase results in selections with more appropriate difficulty: Contrasting Init-correct-CoT with Correct-CoT, the results are 71.4% -> 76.3%.
>
> In the summarization stage, the generated reasoning chains are more comprehensive and refined. Comparing Best-of-N-CoT, Correct-CoT, and Iter-CoT, the results are 73.5% -> 76.3% -> 79.1% (same questions in examples used in the above settings).
>
> We use a **example** to illustrate the differences between Init-wrong-CoT, Correct-CoT and Iter-CoT. Because of word limitations, we have abbreviated the content.
>
> **Question**: At the burger hut, you can buy a burger for $5, french fries for $3, and a soft drink for $3.  If you order a special burger meal, you get all 3 of these food items for $9.50. ..  How much money does Mr. Parker save by buying the 6 meals versus buying the individual food items? (GSM8K Train set item 3446)
>
> **Init-wrong-CoT**:
> First, let's calculate the cost of buying the individual food items for the 2 burger meals for Mr. Parker and his wife.
> ...
> So, Mr. Parker saves $39 by buying the 6 meals instead of buying the individual food items.
>
> **Correct-CoT**:
> The cost of a regular burger meal can be calculated by adding the prices of a burger, fries, and a soft drink, which equals $11
> ...
> To determine the total savings, add the savings on special burger meals and the savings on kid's burger meals, which amounts to $10.
>
> **Iter-CoT**:
> 1. Calculate the cost of a normal burger meal: burger + fries + soft drink = 5+3+3=$11
> ...
> 8. Calculate the cost of 2 special kid's burger meals: 2*5 = $10
> 9. Calculate the total savings: savings on special burger meals + savings on kid's burger meals=6+4=$10.
>
> We can see that Correct-CoT corrects the initial wrong reasoning chain, and then Iter-CoT generates a more comprehensive and detailed reasoning chain based on the Correct-CoT.
>
> And the ‘rising and then falling’ trend, although the trend is not obvious, during multiple rounds of iterations, the range of changes in model performance is not large, and within this limited range we did observe such a trend of first rising and then falling.

---

### Official Review · Reviewer_uFPN · 2023-11-02

**Soundness:** 2 fair
**Presentation:** 3 good
**Contribution:** 1 poor
**Rating:** 3
**Confidence:** 4

**Summary:**

The authors first validate that LLMs possess self-correction ability. They further establish that using more accurate demonstrations in the prompt leads to better results. Building on these two findings, the authors proposed Iter-CoT (iterative bootstrapping in Chain-of-Thoughts prompting), aiming to enhance the performance of large language models on reasoning tasks. This method generates the CoT demonstrations by guiding the LLM to rectify errors and summarize the reasoning chains on questions with appropriate difficulty levels. As such, during the inference, the demonstrations are more accurate compared with other prompting methods. The authors demonstrate the effectiveness of the method on three reasoning tasks on ten datasets, including arithmetic, commonsense, and symbolic reasoning. The results show that using demonstrations generated by Iter-CoT outperforms using traditional prompting methods, and achieves state-of-the-art performance.

**Strengths:**

1. The motivation is strong and supported by two findings, i.e. the self-correction ability of LLMs and the value of revised examples.

2. The Iter-CoT method is straightforward to follow. Experimental results across three categories of reasoning tasks and ten datasets prove the effectiveness of this method.

3. A series of ablation experiments are conducted to investigate the two phases of the method.

**Weaknesses:**

1. The bootstrapping phase relies on GPT-4 to assess examples without labels, which does not address the issue of hallucinations and might lead to misjudgment of correct responses.

2. The summarization phase employs an LLM, which could also lead to hallucinations, especially when dealing with lengthy multi-turn conversation contexts, potentially impairing LLM performance and resulting in suboptimal summarizations. Additionally, including the entire correction process in the summarization phase might lead to misleading summaries.

3. I found the innovativeness of the proposed method is limited, as the verify-and-correct approach has been previously explored in works such as ReAct and self-ask. Additionally, more efficient methods for demonstration sampling, like k-nearest neighbor sampling, are available.

Reference:
Yao, Shunyu, et al. "React: Synergizing reasoning and acting in language models."
Press, Ofir, et al. "Measuring and narrowing the compositionality gap in language models.”
Liu, Jiachang, et al. "What Makes Good In-Context Examples for GPT-3 ?.”

**Questions:**

1. What is the inference cost between Iter-CoT and other baseline methods? And what is the size of the demonstration pool?

2. How many demonstrations are given in the prompt during inference? Is it consistent for all baseline methods?

3. In Figure 6, which specific task does the accuracy pertain to? I assume it relates to tasks with labels, but it is important to note that accurate evaluations on labeled tasks do not necessarily translate to accurate evaluations on tasks without labels, given the varied nature of these tasks.

---

> ### Author Response · Authors · 2023-11-17
>
> **Weaknesses1**:  Bootstrapping hallucinations.
>
> **Answer**:  As shown in Figure 7, we studied the hallucination problem of evaluator and its impact. The conclusion is that the hallucination problem indeed occurs (for example, when GPT-4 is used as an evaluator to evaluate gpt-3.5-turbo, the judgment accuracy is only 88%), but despite the existence of hallucinations, this method still benefits our results, because after sampling, there is almost only one or even none error exemplar (e.g. the gsm8k dataset needs 8 exemplars and 8*88%=7.04, which means that the correct number of examples in the demonstration is about 7 due to the random sampling) in the demonstration, which has a small impact on the results.
>
> **Weaknesses2**: Summary hallucinations.
>
> **Answer**:  You have astutely identified a potential issue. Indeed, during our experimentation, we encountered this issue as well. However, we employed a method to mitigate the impact of model hallucinations during the summarization stage. Specifically, after the model generates the final reasoning chains during the summarization stage, we incorporate an evaluation process. This process employs either labels or an LLM evaluator to reconfirm the correctness of the reasoning chains, retaining only the accurate ones. We will update the explanation of this operation in our method description in the latest version of the paper.
>
> Additionally, despite the possibility of introducing errors by including the record of the correction process in the summarization stage, this step, more often than not, enables the model to avoid repeating past mistakes. This was the intended purpose behind implementing this operation, and its effectiveness is evident in the comparison of Correct CoT and Iter-CoT results in Table 3 (79.2% to 80.9%).
>
> **Weaknesses3**:  Innovativeness of the proposed method is limited.
>
> **Answer**:  We cannot agree that the paper your cited is the verify-and-correct approach.  They actually LLM-Agent works. The most significant distinction between our approach and the two articles you referenced lies in the core concept of our method, which involves learning from errors to generate examples with moderate difficulty and comprehensive, detailed reasoning chains for the model to undergo in-context learning. In contrast,  React and Self-ask use LLM as agent to prioritize planning and decision-making. For instance,  Self-ask decomposes problems into sub-problems (planning), then utilizes a search engine to find answers for each sub-problem (decision-making), and finally summarizes the responses. Additionally,  both of the mentioned methods involve iterative calls to LLMs during the inference stage for each question, resulting in prolonged inference times and higher costs. Conversely, our approach, after constructing a high-quality sample pool, introduces no additional overhead during the inference stage.
>
> In addition, regarding the sampling method for examples, we believe that Iter-CoT, through sampling error-corrected questions, can ensure that the exemplars possess an appropriate level of difficulty—challenging the model while staying within the potential range the model can address. As for the alternative sampling methods you mentioned, we will explore them in the sampling phase from the demonstration pool in our future work.
>
> **Question1**:  Inference cost?
>
> **Answer**:  Our approach incurs the same cost during the inference stage as all other baselines. This is because all additional costs associated with our approach are incurred during the construction phase of the demonstration pool. Once the demonstration pool is constructed, we only require sampling from the pool in the inference stage.
>
> Regarding the size of the demonstration pool, it has both a maximum and a minimum value. The maximum value corresponds to the size of the demonstration pool obtained using Iter-CoT on the entire training set of the dataset. The minimum value corresponds to the number of exemplars required during the inference stage. For example, in our experiments on GSM8K, the minimum value is 8 (refer to Appendix Table 6, which describes the number of exemplars needed for each dataset).
>
> **Question2**:  How many demonstrations?
>
> **Answer**:  Certainly, for the sake of fair comparison, we ensured that the number of examples used during the inference stage aligns with the baselines. The specific numerical details are presented in Appendix Table 6.
>
>
> **Question3**:  Which specific task does the accuracy pertain to?
>
> **Answer**:  As illustrated in the legends of Figure 6, we conducted experiments on three tasks representing arithmetic, common-sense, and symbolic reasoning, denoted as GSM8K, CSQA, and Letter Concatenation, respectively. While these three datasets come with labels, we also performed experiments without utilizing their labels. This allowed us to treat them as tasks without labels, serving as a validation of the accuracy of Iter-CoT(w/o labels).

---

> > ### Comment · Reviewer_uFPN · 2023-11-22
> > **Thanks for your response.**
> >
> > Thank you for your response. I acknowledge that Iter-CoT employs several techniques to mitigate hallucinations in bootstrapping and summarisation and achieves improved results. The authors also mention that an evaluation process was employed after the model generated the final reasoning chains. This significant step, however, is not detailed in the initial paper. Moreover, further analysis on the improvements brought about by this evaluation process would be beneficial. Additionally, to me, the novelty of this paper seems limited.

---

### Official Review · Reviewer_3Dhx · 2023-11-05

**Soundness:** 2 fair
**Presentation:** 3 good
**Contribution:** 2 fair
**Rating:** 6
**Confidence:** 4

**Summary:**

The paper introduces a new way of preparing chain-of-thought demonstration examples by iterative bootstrapping with the training data. To construct the few-shot demonstration pool with reasoning chain from the training set, the framework first generate the initial reasoning chain with Zero-Shot-CoT. It then iteratively refines examples with incorrect answers with the LLM itself, and finally summarize the final reasoning chain once the correct answers are derived. Since the refined require guidance on correctness of the answer, the authors experiment with both using the ground truth label, and using an LLM as evaluator. At inference time, N random examples are sampled from the prepared pool and fix as the few-shot examples. Experiments on 10 datasets covering arithmetic, commonsense and symbolic reasoning showcase that the proposed method achieves new state-of-the-art results.

**Strengths:**

1. The paper propose a new way of constructing few-shot chain-of-thought examples to use for in-context learning. It is the first to apply self-refinement and LLM self-evaluation in construction demonstration examples, and the experimental results are strong with and without ground truth labels.
2. The authors evaluate the method with both proprietary GPT models and open source LLAMA models, and performs well in both cases.
3. The authors conduct thorough ablation to examine different stages of the construction pipeline and impact of LLM evaluator's performance on final accuracy. These results are helpful for understanding the model performance and consistent with the main motivation to use iterative bootstrapping to improve exemplar quality.

**Weaknesses:**

1. It is mentioned that during the inference stage, a random N exemplars are samples as the fixed demonstrations for the entire test set. I have a few questions on this choice:
    1. If only N random exemplars are used for the entire test set for inference, is it still necessary to construct the demonstration pool based on the entire training set? Or what is the use of "pool" here since only N exemplars are used.
    2. Is is possible to have comparison with the baselines use the same set of few-shot examples, just with different reasoning chain annotations. Since the contribution of the paper is mainly on the construction of the reasoning chains, I feel it is better to have the selection of examples consistent for fair comparison. Meanwhile it would be interesting to see if the example selection strategies used in Complex-CoT and Auto-CoT are helpful for Iter-CoT or not.
2. While the experiments are conducted over 10 different datasets, most of them are on the easier side and has some synthetic nature. This also reflects on the results, where the LLMs generally achieve very high scores on these datasets already. It would be better if the authors could evaluate on other more realistic and challenging datasets.

**Questions:**

1. Please see weakness above on the random selection of demonstration examples.

---

> ### Author Response · Authors · 2023-11-17
>
> **Weaknesses1-A**: what is the use of "pool" here since only N exemplars are used.
>
> **Answer**: Your question is highly intriguing, and we have indeed deliberated on this matter. Following the pool's construction on the entire training set, there are various optional methods for sampling examples from the pool, such as random sampling (as employed in our approach), cluster-based sampling, complexity-based sampling, and so on. For random sampling, it is indeed unnecessary to construct the pool on the entire training set. Instead, we can calculate the minimum number of questions N required for the initialization stage based on the model's zero-shot accuracy on the dataset and the number of samples needed during the inference stage. Subsequently, we can randomly sample N questions from the training set for initialization. However, we aspire to explore alternative sampling methods in future work, and these methods would necessitate a larger pool.
>
>
> **Weaknesses1-B**: fair comparison
>
> **Answer**: Firstly, it is essential to clarify that the contribution of Iter-CoT lies not only in its ability to generate more accurate and comprehensive reasoning chains for problems but also in its selection of examples with appropriate difficulty. These factors collectively constitute the source of improvement brought about by Iter-CoT. As mentioned in Section 4.6.1, the generated exemplars feature questions initially answered incorrectly by the model but subsequently corrected through guidance. Such questions pose a certain level of challenge to the model while also offering the potential for correct answers. Therefore, the revision phase can, to the greatest extent possible, ensure the correctness of exemplars while selecting questions of appropriate difficulty.
>
> Therefore, when comparing with other baselines, we do not maintain consistency in the selected questions. Additionally, regarding integrating the selection strategies of Complex-CoT or Auto-CoT into Iter-CoT, as discussed in A1, this is indeed a worthwhile method for exploration, and we can incorporate it into the stage of selecting samples from the pool.
>
>
> **Weaknesses2**:  more realistic and challenging datasets.
>
> **Answer**: Thank you very much for your valuable suggestions. In our future work, we will consider validating our methods on more challenging datasets. Simultaneously, we remain confident that our approach remains effective in more difficult tasks. However, I would like to explain the reasons behind our selection of these 10 datasets. Firstly, we aimed for fairness in comparison with previous studies by choosing datasets consistent with those they used. Secondly, the datasets we selected represent typical examples from three categories of tasks: Arithmetic reasoning, common-sense reasoning, and symbolic reasoning. The experimental results on these tasks provide compelling evidence of the superiority of Iter-CoT.

---

> > ### Comment · Reviewer_3Dhx · 2023-11-22
> >
> > Thanks for the response and additional details. I will keep my rating of weak accept, as I think the comments I have regarding the selection of exemplars and datasets remain open questions. Nevertheless, the findings in the paper is still worth sharing given all the positive results.

---

### Official Review · Reviewer_gmng · 2023-11-05

**Soundness:** 3 good
**Presentation:** 3 good
**Contribution:** 3 good
**Rating:** 5
**Confidence:** 4

**Summary:**

This paper introduces an approach called Iter-CoT that generates accurate rationales through iterative bootstrapping. Initially, Iter-CoT produces a rationale via zero-shot-CoT. If the answer derived from this rationale is incorrect, the rationale is revised until it's correct. Finally, all generated rationales are simplified through summarization. The authors apply this framework to construct few-shot demonstrations of training examples and demonstrate that this method outperforms existing baselines for zero-shot-CoT and even the baselines with ground-truth annotations.

**Strengths:**

1. Overall, the writing is clear and easy to follow. In addition, the organization of the main draft is well-established.
2. Improving the reasoning capability of LLMs is an interesting and important problem. To this end, considering the quality of few-shot demonstrations is a reasonable and well-motivated direction.
3. The proposed method is simple and can be applicable regardless of the types of LLMs. Also, it shows a significant improvement compared to the existing baselines, well-studied in the same problems.

**Weaknesses:**

1. While the improvements are significant, the explanation for such gains is not sufficient. For example, the proposed Iter-CoT outperforms both Manual-CoT and Complex-CoT in Table 1, which use the ground-truth annotations of rationales. Since Iter-CoT is a method to generate correct rationales of unannotated samples, both methods can be considered as the upper bound as they always use the ground-truth rationales. Therefore, it’s not natural that Iter-CoT shows better performance than those ones. What is the source of such improvement?
2. One of the major concerns is an increase in cost due to the iterative usage of LLMs, but there is no discussion regarding this. How many costs are required for Iter-CoT, compared to other methods? Also, it would be better if the authors could have a corresponding discussion in the draft.
3. While the proposed method is applied to generate rationales of the training examples for few-shot demonstrations, it can be applicable during the inference. Also, the framework of bootstrapping the rationale has been widely explored [1,2]. Hence, I’m wondering if the improvement can be enlarged when Iter-CoT is also applied during inference.

### Minor

1. It would be better to change the order of the presented method in Table 3, with the decreasing order for better presentation.
2. Also, as the authors presented in the Appendix, STaR is highly relevant to the proposed Iter-CoT. It seems to be better to add this baseline to the main table if possible.
3. Regarding Figure 6, the authors mention that “we utilize the best exemplars in this section”. What is the meaning of best exemplars? Also, how do you select them? Do you utilize another validation set to choose them?

[1] Madaan et al., Self-Refine: Iterative Refinement with Self-Feedback., NeurIPS 23
[2] Shinn et al., Reflexion: Language Agents with Verbal Reinforcement Learning., NeurIPS23

**Questions:**

Please answer the questions and concerns in weaknesses. If the responses are enough to address my concerns, I will raise my score accordingly.

---

> ### Author Response · Authors · 2023-11-17
>
> **Weaknesses1**: What is the source of such improvement?
>
> **Answer**: We cannot agree that Manual-Cot and Complex-CoT represent the upper bounds. Providing correct rationales is not sufficient, because the value of an exemplar is derived not solely from the accuracy of its reasoning chains but also from the comprehensive and difficulty level of its reasoning chains. Manual-CoT ensures the correctness of reasoning chains in exemplars but does not control the difficulty level of the problems within them. On the other hand, Complex-CoT indeed ensures the correctness of reasoning chains and selects the most complex exemplars. However, as illustrated in Figure 1, it tends to choose exemplars that are overly complex, leading to suboptimal performance on simpler problems.
>
> Different from the two methods, Iter-CoT (1) ensures the correctness of reasoning chains through error correction (We have used some methods to reduce the impact of model hallucinations, so we disregard model illusions in this discussion); (2) ensures appropriate difficulty levels by selecting corrected exemplars, resulting in superior performance across a spectrum of difficulty levels; (3) enhances the comprehensiveness and thoroughness of the reasoning chains through summerization step. That's the source of its improvement. We elaborate on these aspects in Section 4.6.1 and Section 4.6.2 of our paper.
>
> **Weaknesses2**: Cost due to the iterative usage of LLMs
>
> **Answer**: The table below illustrates the numbers associated with the minimum cost incurred by Iter-CoT in generating examples, considering the model's zero-shot accuracy on the dataset and the number of examples used during inference.
>
> | Iteration | Zero-shot-CoT Accuracy | First Correction Success Rate | Examples Used | Samples Sampled | Total Model Calls |
> |-------|--------------------|-------------------------------|---------------|----------------|------------------|
> |    1      |       54.7%        |            27.3%              |       8       |        65      |       103        |
>
> In this example, assuming ChatGPT (old version) achieves a zero-shot accuracy of 54.7% on gsm8k, and based on preliminary experimental results, the success rate of the first correction is estimated to be half of the zero-shot accuracy, i.e., 27.3%. During the inference phase, eight examples are required.
>
> To determine the number of samples to be sampled from the training set, we calculate it as approximately 65, using the formula 8 divided by 27.3% divided by (1 - 54.7%).
>
> The total number of calls to the model is 103. This includes zero-shot inference on 65 samples, correcting approximately 30 errors (calculated as 65 multiplied by (1 - 54.7%), which is approximately 30), and summarizing the successful corrections (calculated as 30 multiplied by 27.3%, which is approximately 8).
>
> It is important to note that our method incurs additional LLM calls during example generation compared to other methods, while the cost during inference is comparable to other methods.
>
> **Weaknesses3**:  Can be applicable during the inference.
>
> **Answer**: You are correct. As shown in Figure 3, allowing the model to re-answer questions it originally answered incorrectly can improve the model's accuracy on the dataset. Therefore, Iter-CoT naturally would be effective during the inference stage. However, due to cost considerations, we do not recommend its application during the inference stage.
>
> **Minor1**:
>
> **Answer**: Thank you very much for your suggestion. We will make corresponding adjustments in the updated paper.
>
> **Minor2**:
>
> **Answer**: In fact, we have presented a comparison of Iter-CoT with STAR CoT on GSM8K in Appendix B.4. However, we sincerely apologize for the oversight, as the STAR CoT results were based on experiments conducted with the previous version of chatgpt. We have now re-conducted the experiment using the new version of GPT-3.5-turbo, and the updated results are as follows: STAR CoT achieved 76.3% results on GSM8K. We will incorporate these findings into the latest version of the paper, considering STAR CoT as our baseline and conducting experiments on other datasets as well.
>
> **Minor3**:
>
> **Answer**: In Figure 6, we conducted experiments to investigate the impact of iterations in the bootstrapping stage on the results. "Best exemplars" refer to exemplars generated during multiple iterations that demonstrated the best performance on the test set. For instance, on gsm8k, exemplars obtained after two iterations exhibited the best performance on the test set. Therefore, exemplars generated in the second iteration are considered the "best exemplars."